# Ultralow-temperature-driven water-based sorption refrigeration enabled by low-cost zeolite-like porous aluminophosphate

Zhangli Liu[1,5], Jiaxing Xu[2,5], Min Xu [1,3,4,5✉], Caifeng Huang [1,4], Ruzhu Wang [2], Tingxian Li [2✉] & Xiulan Huai [1,3,4✉]

Thermally driven water-based sorption refrigeration is considered a promising strategy to realize near-zero-carbon cooling applications by addressing the urgent global climate challenge caused by conventional chlorofluorocarbon (CFC) refrigerants. However, developing cost-effective and high-performance water-sorption porous materials driven by low-temperature thermal energy is still a significant challenge. Here, we propose a zeolite-like aluminophosphate with SFO topology (EMM-8) for water-sorption-driven refrigeration. The EMM-8 is characterized by 12-membered ring channels with large accessible pore volume and exhibits high water uptake of 0.28 g·g$^{-1}$ at $P/P_0 = 0.2$, low-temperature regeneration of 65 °C, fast adsorption kinetics, remarkable hydrothermal stability, and scalable fabrication. Importantly, the water-sorption-based chiller with EMM-8 shows the potential of achieving a record coefficient of performance (COP) of 0.85 at an ultralow-driven temperature of 63 °C. The working performance makes EMM-8 a practical alternative to realize high-efficient ultralow-temperature-driven refrigeration.

[1] Institute of Engineering Thermophysics, Chinese Academy of Sciences, Beijing 100190, China. [2] Research Center of Solar Power & Refrigeration, School of Mechanical Engineering, Shanghai Jiao Tong University, Shanghai 200240, China. [3] University of Chinese Academy of Sciences, Beijing 100049, China. [4] Nanjing Institute of Future Energy System, Nanjing 211135, China. [5] These authors contributed equally: Zhangli Liu, Jiaxing Xu, Min Xu. ✉email: xumin@iet.cn; Litx@sjtu.edu.cn; hxl@iet.cn

Energy demands for heating and cooling account for more than 50% of global final energy consumption and are expected to maintain a rapidly increasing rate in the future decades, especially for cooling purposes[1–4]. Accordingly, carbon dioxide (CO$_2$) emissions related to heating and cooling have grown significantly since most of the heat and cold is produced by fossil fuels at present[1,5,6]. Therefore, it is urgent to shift heating and cooling to be a more low-carbon process. Water-based adsorption heat pumps (AHPs) and adsorption chillers (ADCs) (see the working principle in Supplementary Fig. 1 and Note 1), driven by sustainable low-grade thermal energy, are considered as potential alternatives to mitigate climate change and fuels shortage[7–10]. Moreover, the utilization of natural refrigerants of water in adsorption systems is more environmental-friendly than fluorocarbon type refrigerants in vapor compression systems. Thus, near-zero-carbon-emission adsorption heating and cooling have triggered a new round of worldwide research to meet the fast-growing energy demands and face the vast challenges of global warming and environmental pollution in recent decades.

For water-sorption-driven heat pumps and chillers, adsorbents are the critical factor in determining the performance of systems[11]. The perfect water adsorbents should exhibit the following characters: (i) S-shaped water-adsorption isotherms and superior water uptakes; (ii) low regeneration temperatures; (iii) excellent hydrothermal stabilities; (iv) fast water adsorption and desorption kinetics; (v) facile and low-cost synthesis. Traditional water adsorbents, such as silica gels[12,13] and zeolites[14,15], have been extensively investigated for adsorption heating and cooling systems. Despite they show advantages of acceptable price and desirable stabilities, obvious drawbacks are identified, including high regeneration temperatures (>150 °C for zeolites) and low working capacities, resulting in poor energy efficiency and low power density for adsorption systems. Alternatively, metal–organic frameworks (MOFs) with highly tunable pore structures and hydrophilic/hydrophobic functional groups, are considered as new effective water adsorbents[16–18]. During the last decades, lots of new MOFs with high water adsorption capacities have been reported, wherein Zr- and Al-based MOFs, such as MOF-801[19], MOF-303[20], MIP-200[21], MIL-160[22], CAU-23[23], Zr-furm[24], and KMF-1[25], gain much attention due to their notably hydrothermal stability. Among them, KMF-1 and MIP-200 show high energy efficiency for both heating and cooling applications. The lab-scale ADC prototypes using MOFs have also been recently demonstrated[26,27]. However, although many attempts to realize commercial-scale production are reported[28,29], challenges from expensive raw materials and low-scalability in synthesis make MOFs unaffordable in this stage[7]. Consequently, it is exceedingly essential to exploit novel water adsorbents to achieve a breakthrough in both low-cost and high-performance.

Aluminophosphate molecular sieves (AlPOs), a family of zeolite-derivatives-based porous materials, show comparable low cost in synthesis and reasonable market availability for AHP and ADC applications. For example, SAPO-34, usually be used as a catalyst for menthol to olefin reaction, has been investigated for more than two decades as a promising water adsorbent[30,31]. Several years ago, the company of Mitsubishi Plastics from Japan developed three types of aluminophosphates (namely AQSOA-Z01, AQSOA-Z02 and AQSOA-Z05), representing a considerable advancement for ADC applications owing to their outstanding performances of high water uptake and low regeneration temperature (< 90 °C)[32–34]. The attractive performances of lab-scale and commercial-scale ADCs using AQSOA-Z02 have been demonstrated[35,36]. In 2012, Ristić et al.[37] reported the first test of APO-Tric for water-sorption applications. This microporous aluminophosphate shows S-shaped water-adsorption isotherm and excellent water-sorption-based heat storage performance.

The water-adsorption mechanism of this AlPO was revealed that the formation of highly ordered H-bond water cluster is followed by the initial water sorption on Al, previously coordinated to two F in the APO-Tric. Recently, a small-pore aluminophosphate AlPO-LTA was also reported as a promising candidate for water-sorption-based heat pump and thermal energy storage[38]. Moreover, aluminophosphates deliver another advantage of considerable industrial-production capacity. For instance, a SAPO-34 manufacturing plant with the production capacity of 5000 tons per year in Dalian was built in 2018[39]. Unfortunately, the AlPOs reported so far cannot hold a candle to the best current MOFs, such as Co-CUK-1 and KMF-1, especially in terms of the performance of ultra-low-temperature-driven chillers. More investigations on screening and design of AlPOs for water-sorption-based heating and cooling applications are urgently needed.

Among all reported AlPOs, an SFO-type aluminophosphate of EMM-8, fabricated by Cao et al. in 2007[40], shows potentially suitable pore properties and hydrophilicity. The accessible pore volume of EMM-8 (19.57%) is larger than that of usually used SAPO-34 (17.27%) and AlPO-5 (14.07%) according to the database of zeolite structures, suggesting potential better water-adsorption capacity. Also, the diameter of a 12-ring window for EMM-8 is ~0.7 nm, larger than that of 8-ring openings for SAPO-34 and AlPO-LTA (~0.4 nm), indicating weaker hydrophilicity and easy desorption of water molecules at a lower driven temperature. Remarkably, this EMM-8 can be synthesized based on very cheap and common raw materials, pseudo-boehmite and phosphoric acid. A wildly used and scalable compound, 4-dimethylaminopyridine (4-DMAPy), is exploited as a structure-directing agent (SDA), making the synthesis of EMM-8 further low-cost. Although the structure of EMM-8 was reported in 2007[40], its potential in water-based applications has not been exploited to date as a result of insufficient investigations on materials synthesis, characterizations, and rare evaluations on gas sorption performance.

In this work, we report, to the best of our knowledge, first-ever use of EMM-8 as an efficient water adsorbent for ultra-low-temperature-driven heating and cooling. We thoroughly characterize it towards water-adsorption application and achieve a 100 g-scale fabrication. By comprehensive water sorption evaluation, we find the EMM-8 represents step-wise water adsorption, facile desorption at low temperature, and desirable hydrothermal stability. More importantly, EMM-8 shows a notably high COP for cooling and heating at lower driven temperatures than those of the best current state-of-the-art materials, indicating a promising application potential for the exploitation of yet mostly unused low-grade heat. To elaborate the mechanism of high efficiencies at ultralow temperatures, we carried out the theoretical simulation together with experimental measurements, uncovering the high COP originates from a considerably low isosteric heat of adsorption contributed by the hydrogen-bonded water clusters within the pores of EMM-8. This work provides a more effective water adsorbent for realizing affordable, scalable, and high-performance adsorption heating and cooling.

## Results

**Characterization and scalable synthesis.** We prepared EMM-8 with hydrothermal synthesis according to the method reported by Afeworki et al.[41]. X-ray diffraction (XRD) measurements (Fig. 1a) confirms that the prepared aluminophosphate possesses the SFO topological structure. As shown in Fig. 1b and Supplementary Fig. 2, the basic building units of EMM-8 are double four-rings, which form a two-dimensional framework with 12-ring and 8-ring channels. The $^{27}$Al and $^{31}$P magic-angle spinning (MAS) nuclear magnetic resonance (NMR) spectrums (Fig. 1c) and

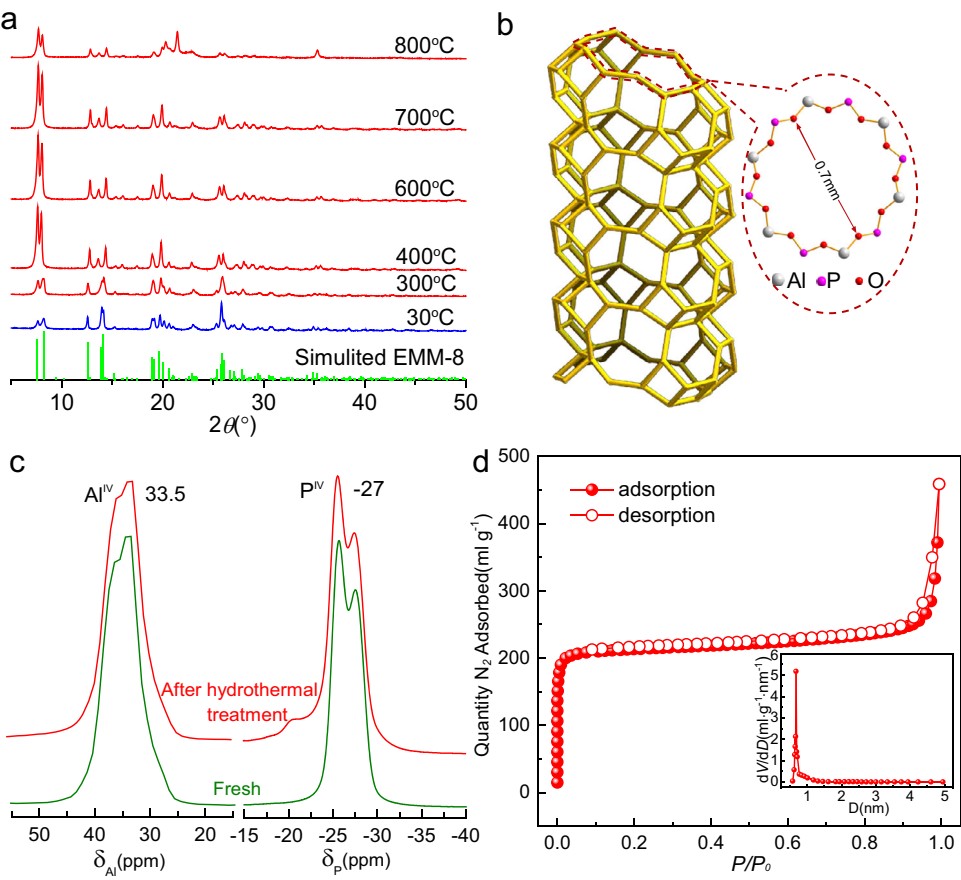

**Fig. 1 Structure, texture, and hydrothermal stability of EMM-8. a** Temperature-dependent in-situ powder X-ray diffraction of EMM-8. The heating rate was 13 °C min⁻¹, and the holding time was 45 min. **b** Structure schematic of the 12-ring channel of EMM-8, which is obtained through its crystal structure determined by Cao et al.[40]. **c** 27Al and 31P magic-angle spinning (MAS) nuclear magnetic resonance (NMR) spectra spectra of EMM-8 for the fresh calcined sample (green) and the same sample after hydrothermal treatment (red). **d** Nitrogen sorption isotherms of the calcined EMM-8 taken at 77 K, the inset image gives the pore size distribution of the sample. Source data are provided as a Source Data file.

X-ray photoelectron spectroscopy (XPS) measurements (Supplementary Fig. 3) of the calcined EMM-8 also confirm the tetrahedrally coordinated framework with Al and P atoms. Nitrogen adsorption results also show the EMM-8 has a high BET surface area of 879.63 m² g⁻¹ and a large pore volume of 0.59 cm³ g⁻¹ (Fig. 1d and Supplementary Table 2). The EMM-8 shows high porosity of 0.45 and micropore volume of 0.3 cm³ g⁻¹, comparable to that of the best current water-adsorption sorbents (Supplementary Table 5). Moreover, we find a concentrated pore diameter distribution with an average micropore size of 0.7 nm, corresponding to the size of 12-ring windows. The nitrogen adsorption isotherm also confirms the presence of the mesopores, whose volume is as high as 0.29 cm³ g⁻¹ that is close to that of micropores.

Scanning electron microscopy (SEM) and TEM images show two-edged sword-shaped and highly faceted EMM-8 particles (Fig. 2a–c), indicating the good crystallinity and high purity of synthesized samples. High-resolution TEM image (Fig. 2d) of the material shows clear lattice fringes, and selective area electron diffraction (SAED) pattern (Fig. 2e) clearly suggests its single-crystal characteristics. Moreover, we find the zeolite-like crystals have uniform shapes in length (1–3 μm) and width (200–300 nm). SEM and atomic force microscope (AFM) observations (Fig. 2b, f–h) reveal that the thicknesses of the EMM-8 nanoplates are distributed between 20 and 120 nm. The gap between the adjacent nanoplates contributes to the presence of mesopores. It is expected that the nano-sized

plate-like structures with abundant intervals offer a high potential to achieve a faster transfer of water molecules within interparticle spaces and micropores.

The thermal conductivity of EMM-8 was measured by the laser flash method at various temperatures. The EMM-8 pellets with packing density of 0.93 g ml⁻¹ shows the thermal conductivities of 0.082–0.37 W (m K)⁻¹ at 25–100 °C (Supplementary Fig. 4a). These values are close to that of adsorbent packed beds of commercial silica gel (0.08–0.15 W (m K)⁻¹)[42] and the MOFs (0.06–0.12 W (m K)⁻¹)[43]. The specific heat capacity of dry EMM-8 was also measured via DSC method at various temperatures. This material represents the specific heat capacities of 0.63–0.93 J (g K)⁻¹ (Supplementary Fig. 4b), comparable to that of the most of the porous materials, which is in the range from 0.6 to 1.1 J (g K)⁻¹[17].

For the water-sorption-based heating and cooling applications, it is crucial that the adsorbent has hydrothermally stable pores. Temperature-dependent powder XRD (Fig. 1a) and thermogravimetric analysis (TGA) (Supplementary Fig. 5) reveal the total removal of the SDA at ~400 °C and demonstrate the desirable thermal stability of the synthesized EMM-8 up to 700 °C, which is slightly lower than that of AlPO-LTA and AlPO-Tric, but much better than that of MOFs. Also, we performed the hydrothermal stability tests by soaking the calcined EMM-8 in boiling water for 24 h, and the results point out that the obtained sorbent can maintain its crystallinity, framework structure, and microporosity, suggesting its stability

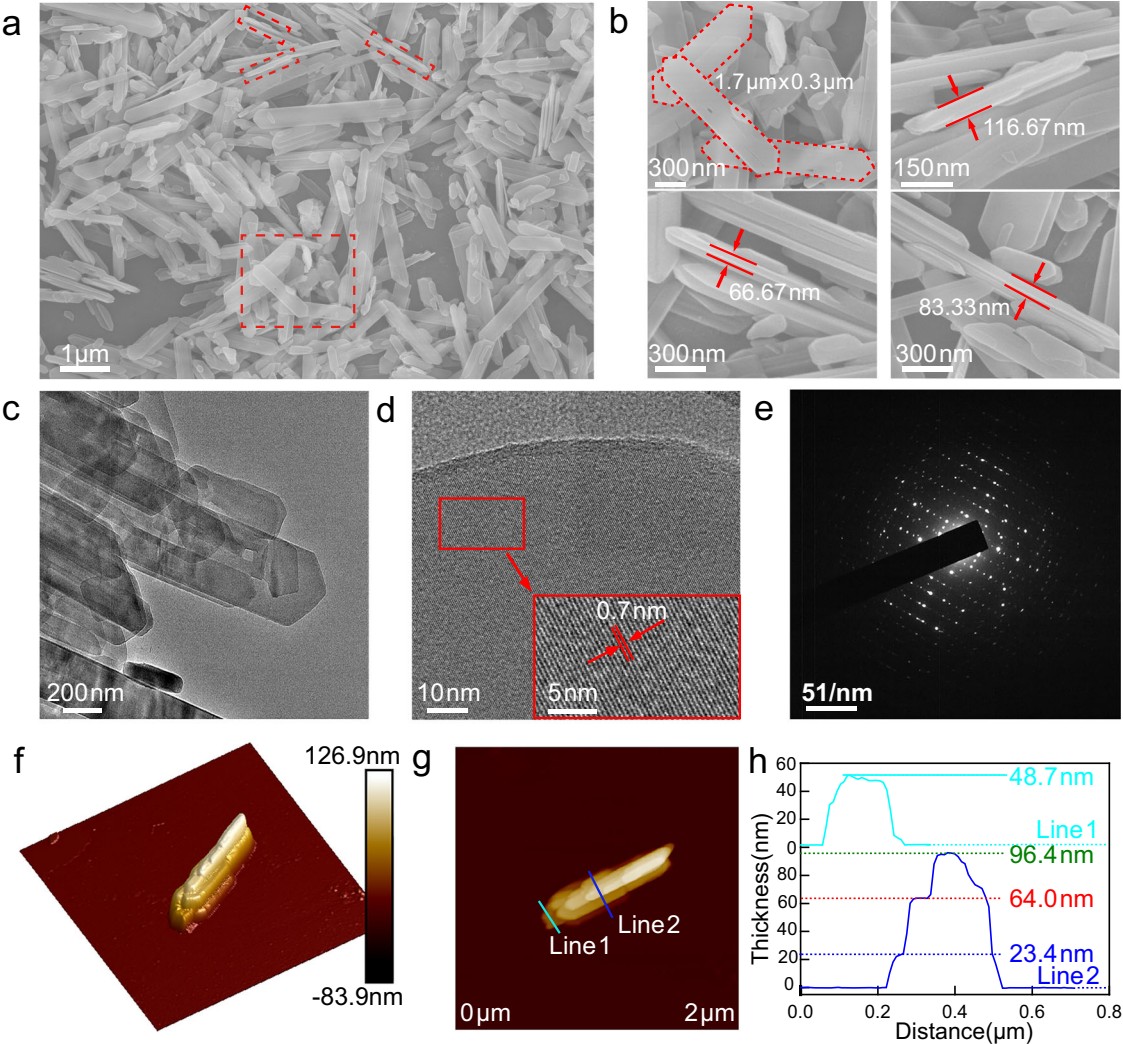

**Fig. 2 Morphologic characterization of the as-obtained EMM-8. a**, **b** Scanning electron microscopy images of the EMM-8 with different magnification. The red dotted areas in **a** were magnified and observed, as shown in **b**. **c–e** Transmission electron microscopy images of the EMM-8 with different magnification and its selected area electron diffraction (SAED) patterns. **f–h** Atomic force microscope images of the EMM-8. Source data are provided as a Source Data file.

under harsh working conditions. This conclusion was further evidenced by the NMR spectrum (Fig. 1c), XRD patterns, and $N_2$ adsorption isotherms (Supplementary Fig. 6).

To build a real-life full-scale ADC device, it is imperative that the sample synthesis process of water adsorbents should be easily scaled up. Thus, the hundred-gram-scale production of this material is performed by a scaled-up reactor in this work. As shown in Supplementary Fig. 7a, we use a 2 L Teflon vessel to replace the 50 mL one in synthesis and introduce 40 times the number of raw materials in it. As a result, about 100 g EMM-8 with good crystallinity is obtained, suggesting the robust fabrication of this aluminophosphate (Supplementary Fig. 7b). Importantly, it is found that there is no significant difference between crystals of EMM-8 synthesized by 50-mL and 2-L Teflon vessels, evidenced by SEM images, XRD patterns, and $N_2$ adsorption isotherms (Supplementary Fig. 8a–c). Moreover, it is worth mentioning that EMM-8 is easily shaped as pellets by pressing without any binders (Supplementary Fig. 8d).

**Water-adsorption evaluation.** The water-adsorption isotherms of EMM-8 were measured by a gravimetric adsorption analyzer at three different temperatures (25, 40, and 50 °C) (Fig. 3a). The results represent a perfect S-shaped water-adsorption isotherm with a step-wise water uptake in the extremely narrow relative pressure range of $P/P_0 = 0.15$–0.17, indicative of the presence of uniform micropores. EMM-8 exhibits a relatively high water-adsorption capacity of 0.283 $g_{H2O} g_{sorbent}^{-1}$ at 25 °C and $P/P_0 = 0.2$, comparable to that of the selected state-of-the-art MOFs and AlPOs (Supplementary Fig. 9a, b), including SAPO-34, AlPO-Tric, MOF-801, CAU-10, COF-TpPa[44], and Co-CUK-1[45]. A small hysteresis is observed in the desorption branch in the narrow range of $P/P_0 = 0.1$–0.15 (Supplementary Fig. 9c), which may be attributed to the presence of the mesoporous.

A calculated average isosteric enthalpy of water adsorption was determined to be 46.76 kJ mol$^{-1}$ (Fig. 3b) by using the Clausius–Clapeyron equation based on the data of water-adsorption isotherms at three different temperatures. We found this value of EMM-8 is lower than that of many other benchmark sorbents (Fig. 3b and Supplementary Table 5) and merely higher than the evaporation enthalpy of water (44 kJ mol$^{-1}$), indicating a relatively lower energy consumption for water desorption and thus higher COP for sorption-based heating or cooling system. Unlike the strong electrostatic interaction between the framework and

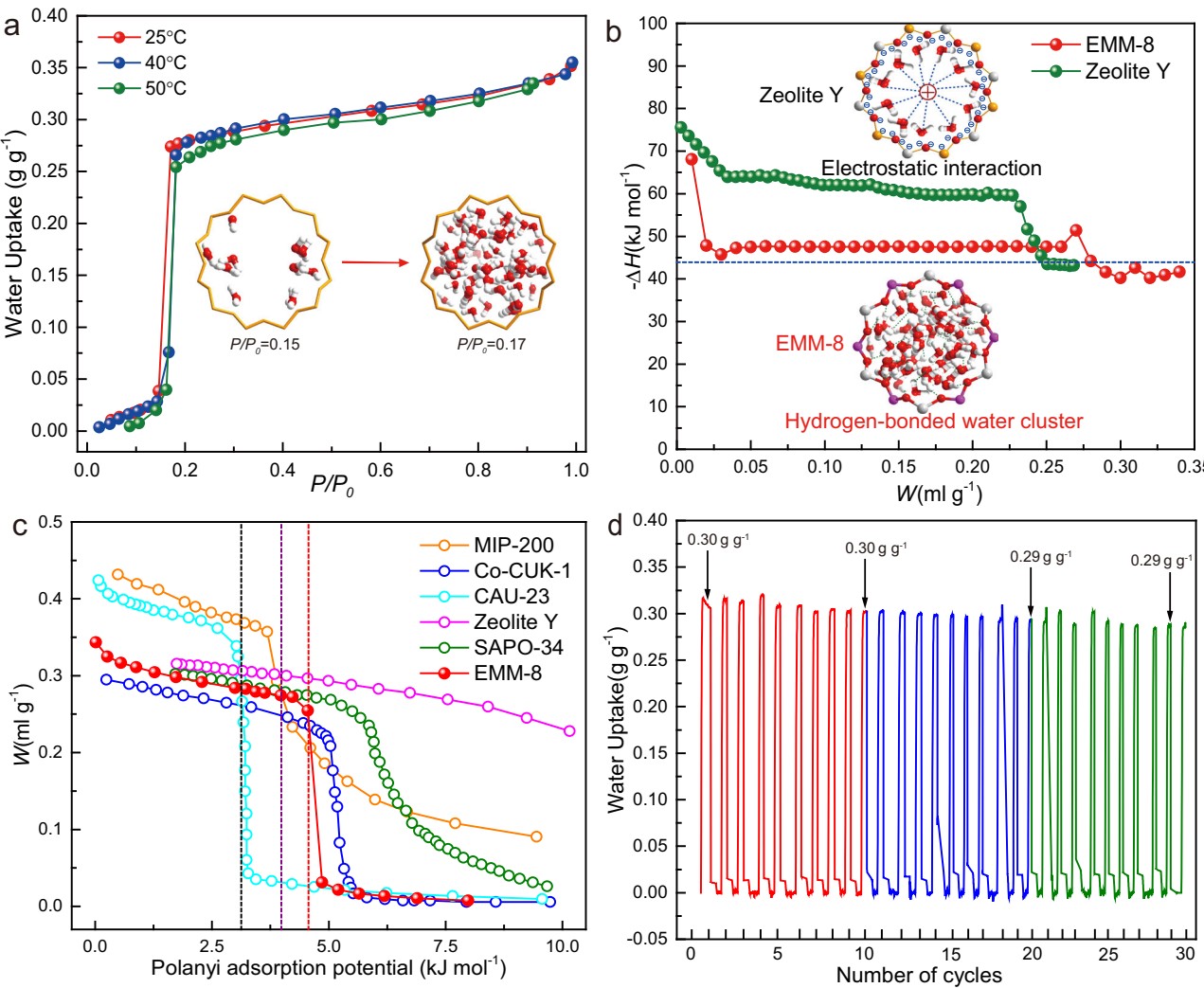

**Fig. 3 Water-adsorption properties of EMM-8. a** Water-adsorption isotherms recorded at three different temperatures. The structure in the inset represents the schematic diagram of step-wise water uptake between $P/P_0 = 0.15$ and $P/P_0 = 0.17$. The yellow structure refers to 12-ring channel with adsorbed water molecules. Red and white spheres refer to oxygen atoms and hydrogen atoms, respectively. **b** The calculated adsorption enthalpy as a function of water loading. The blue dash line represents the water evaporation enthalpy. The structure in the inset represents the interaction between different structural channels and water molecules. The closed rings refer to the channels of Zeolite Y and EMM-8. Red and white spheres refer to oxygen atoms and hydrogen atoms of water molecules, respectively. **c** Characteristic curves determined by adsorption isotherm at 30 °C of EMM-8 and other reference materials. The dashed lines represent the optimal adsorption potential for cooling to 10 °C from 30 °C (black), cooling to 5 °C from 30 °C (purple), and heating to 45 °C from 15 °C (red). **d** Water-adsorption/desorption cycling performance of EMM-8. Test conditions: adsorption at 30 °C with 30% RH, and desorption at 110 °C under vacuum. Source data are provided as a Source Data file.

water molecules for conventional zeolite, the zeotypic EMM-8 with electrical neutrality endows a weak hydrogen-bonded network of water molecules within micropores, leading to lower adsorption enthalpy, which will be detailed in the following section. The characteristic curves defining the relations of adsorption potential and water uptake[16] are determined using adsorption isotherms of EMM-8 and other reference materials, as shown in Fig. 3c. In the context of meeting the typical operating conditions for both refrigeration and heat pump applications, EMM-8 exhibits a significantly lower adsorption potential than conventionally used zeolite, SAPO-34, and Co-CUK-1, indicating the lower required driving temperature.

In order to further confirm the suitability of the material for applications on ADC and AHP, cycling stability tests were carried out under the condition of regeneration temperature of 110 °C and adsorption temperature of 30 °C. The results reveal no significant loss of the water uptake for EMM-8 over 30 adsorption/desorption cycles (Fig. 3d).

To further explore the adsorption mechanism at the microscopic level, the water-adsorption isotherm of EMM-8 was calculated by using Grand Canonical Monte Carlo (GCMC) simulations at 30 °C. The obtained result matches the experimental adsorption isotherm well, as shown in Supplementary Fig. 10. At a low relative pressure of $P/P_0 = 0.01$, only a small amount of water molecules enter into the 12-ring channels, associated with a high adsorption enthalpy (71 kJ mol⁻¹, Supplementary Fig. 11). The water molecules are preferentially coordinated to aluminum atoms on the pore wall (Fig. 4a), resulting in the change of the aluminum atom from tetrahedral coordination to octahedral coordination. This phenomenon is very common for water adsorption on AlPOs[39,46–48] and consistent with the ²⁷Al NMR experimental findings (Supplementary Fig. 12a). When the relative water vapor pressure is above 0.25, there is a sudden increase in the water-adsorption capacity and the water molecules tend to fill the entire 12-ring 1D pore openings of EMM-8, as shown in Fig. 4b, c. It also can be

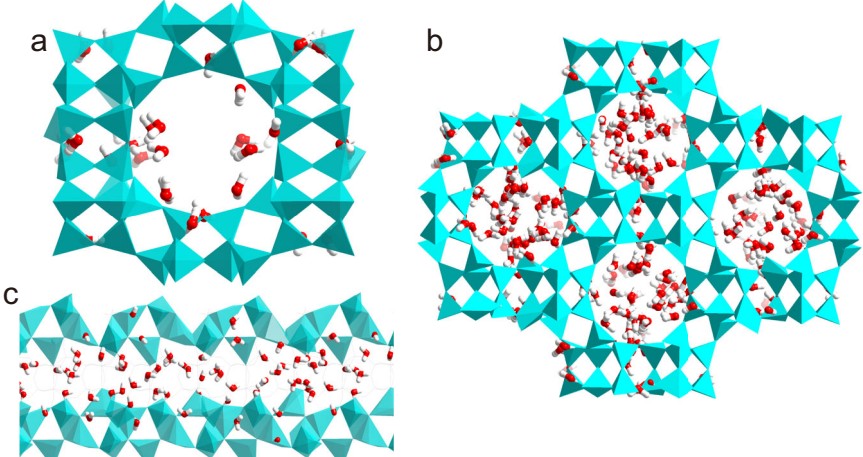

**Fig. 4 Grand canonical Monte Carlo (GCMC) simulation results of water adsorption. a** The adsorbed water molecules coordinating to aluminum atoms in EMM-8 at low pressure ($P/P_0 = 0.01$). **b** Aggregation of hydrogen-bonded water molecules in the 12-ring channel at $P/P_0 = 0.3$ from the top view. **c** The cross-section of the water-loaded channel. The cyan structure refers to the EMM-8 framework. The red and white spheres refer to oxygen atoms and hydrogen atoms of water molecules, respectively.

seen that a hydrogen bond network of water molecules is formed with a total number of hydrogen bonds per water molecule of 2.9 (Supplementary Fig. 13), which is close to that of bulk water. This observation is supported by the $^1H$ NMR spectra (Supplementary Fig. 12b), which shows the low-field shift of the water signal, indicating the presence of the hydrogen-bonded water molecules. This low-energy water–water molecular interaction suggests a low enthalpy of adsorption (40–50 kJ mol$^{-1}$, Supplementary Fig. 11) and thus results in the facile regeneration of EMM-8.

The pore-filling mechanism of water uptake mentioned above is also observed in other AlPOs, such as AlPO-LTA, AlPO-Tric, and AlPO-18, suggesting the similar stepwise isotherms of these aluminophosphates (Supplementary Fig. 9b). The exception is SAPO-34. The incorporation of Si, leading to the formation of highly acidic bridging OH groups as rather strong adsorption centers for water molecules, are associated with high water uptake at low relative pressure. However, EMM-8 shows lower heat of adsorption than other reported AlPOs, as shown Supplementary Table 5. This is mainly due to the weaker water–framework interactions and the weaker interaction between the water molecules contributed by the larger cavity of EMM-8 with 12-ring openings than that of AlPO-LTA and AlPO-Tric with 8-membered rings, which is illustrated by Demontis et al.[49]. We could reveal the structure-property relationship, where appropriately weaker water-adsorption sites and properly larger pore sizes of EMM-8 result in the low adsorption enthalpy, providing the guidance to design the porous materials for water-sorption applications.

**ADC performance evaluation.** Materials-based COPs for cooling, which are usually used to assess the energy efficiency of the ADC devices, are evaluated according to the methodology by de Lange et al.[18]. (Supplementary Note 2) at different boundary temperatures for evaporation ($T_{ev}$), condensation ($T_{con}$), and regeneration ($T_{des}$). Figure 5a shows the calculated maximum COP$_C$ for EMM-8 and its corresponding driving temperature for a specific refrigeration condition, i.e., $T_{ev} = 5$ °C and $T_{con} = 30$ °C. The results indicate that EMM-8 exhibits an exceptionally high COP$_C$ of 0.85 at the ultralow driving temperature of 63 °C, remarkably exceeding the existing state-of-the-art materials, including the recently reported best-in-class adsorbents, such as AlPO-LTA (0.75), MIP-200 (0.78), KFM-1 (0.75), and

Co-CUK-1 (0.83) (Supplementary Table 4). Additionally, the very high thermal efficiencies and working capacities on cooling retain for a wide range of evaporation temperatures, as illustrated in Supplementary Figs. 14a and 15. More importantly, the required driving temperatures, achieving the highest COP and working capacity of EMM-8, are lower than those of reference materials by 5–15 °C, as shown in Fig. 5a, b, which is significantly meaningful for the efficient exploitation of ultralow-grade thermal energy. The complementary metric, specific energy capacity, i.e., the provided cooling capacity from the evaporator in one cooling cycle ($Q_{ev}$), is evaluated and compared with that of other benchmark materials (Supplementary Fig. 16). The excellent volumetric specific energy capacity for EMM-8 is also found even at a low regeneration temperature of 65 °C (Fig. 5c and Supplementary Table 5), which is another indication of the extraordinary cooling performance of EMM-8 at ultralow driving temperatures.

Moreover, under a standard heat pump condition, i.e., $T_{ev} = 15$ °C and $T_{con} = 45$ °C, EMM-8 also gives a very high COP$_H$ of 1.75 at a driven temperature of 82 °C (Fig. 5d and Supplementary Fig. 14b). The comparison of this value between EMM-8 and other excellent materials confirms that EMM-8 outperforms both other AlPOs and most MOFs for heat pump applications. This value is also as high as that of the best water adsorbents reported so far, i.e., Co-CUK-1 (1.77) and KMF-1 (1.74). However, in comparison with Co-CUK-1, EMM-8 shows higher volumetric working capacities at lower desorption temperatures (Fig. 5e). Therefore, the obtained high cooling and heating efficiencies suggest that EMM-8 is, to the best of our knowledge, one of the best water adsorbents to date for realizing ultra-low-temperature-driven ADC and AHP applications.

Alongside COP, the specific cooling power (SCP) of the water adsorbent is also a key performance indicator that dictates the power density of ADC devices. EMM-8 was shaped via pressing, crushing, sieving, and then deposited on a flat-plate adhesive plate (Supplementary Fig. 18b) to assess its dynamic sorption performances under varying ADC operating conditions (Supplementary Table 7). The water desorption profiles of EMM-8 show the fast desorption rates under the operating conditions of $T_{con} = 30$ °C, and $T_{des} = 65$ °C (Fig. 5f), demonstrating its superiority in the deep utilization of ultra-low-temperature heat. We calculated the diffusion coefficient of water vapor within the particles base on the water-adsorption kinetic curves

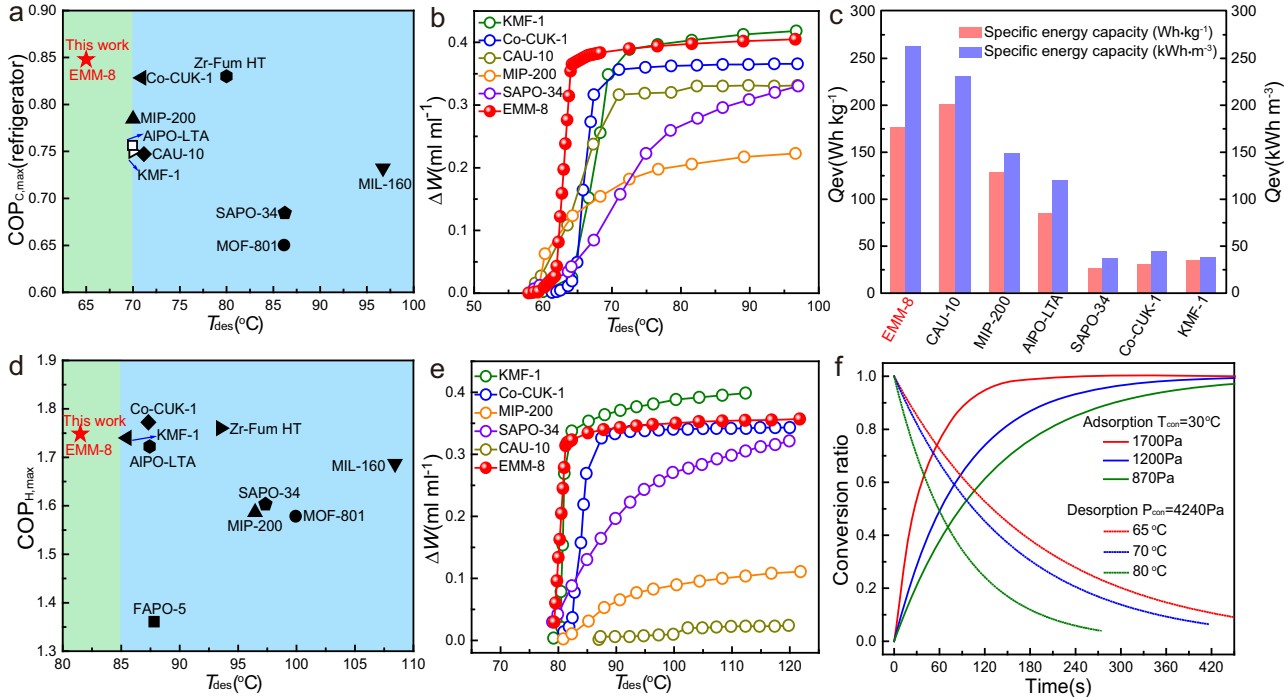

**Fig. 5 Performance evaluation of EMM-8 in comparison with other adsorbents. a** Comparison of maximum coefficient of performance (COP) values and their corresponding driven temperature for cooling of EMM-8 and reference materials. Refrigeration conditions used were $T_{ev} = 5\,°C$ and $T_{con} = 30\,°C$. **b** Plots of the volumetric working capacities under the varying regeneration temperatures for adsorption chiller (ADC) conditions ($T_{ev} = 5\,°C$ and $T_{con} = 30\,°C$). **c** Specific energy capacities for EMM-8 and benchmark adsorbents expressed in gravimetric and volumetric scales ($T_{ev} = 10\,°C$, $T_{con} = 30\,°C$, and $T_{des} = 65\,°C$). **d** Maximum coefficient of performance (COP) values versus the driven temperature for heat pump application. The conditions used for calculation were $T_{ev} = 15\,°C$ and $T_{con} = 45\,°C$. **e** Plots of the volumetric working capacities under the varying regeneration temperatures for adsorption heat pump (AHP) conditions ($T_{ev} = 15\,°C$ and $T_{con} = 45\,°C$). **f** Dimensionless water uptake curves during adsorption and desorption run under varying conditions. The size of EMM-8 grain is 0.45–0.6 mm. Green and blue areas in **a** and **d** respectively refer to the required driven temperatures ranges for EMM-8 and reference materials to achieve the maximum coefficient of performance (COP). Source data are provided as a Source Data file.

(Supplementary Note 3), as shown in Supplementary Table 8. According to the evaluations of SCPs based on kinetic measurements (Supplementary Note 3), EMM-8 has significantly high power density in ADCs operated at low evaporator temperature (5–10 °C) and low driven temperatures (65–80 °C), as seen in Supplementary Tables 9 and 10. Typically, even under the rather harsh working conditions ($T_{ev} = 5\,°C$, $T_{con} = 30\,°C$, and $T_{des} = 65\,°C$), a superior high $SCP_{max}$ of $2.22\,kW\,kg_{sorbents}^{-1}$ and $SCP_{80\%}$ of $1.1\,kW\,kg_{sorbents}^{-1}$ for this material with the size of 0.45–0.6 mm is achieved. The $SCP_{80\%}$ value of EMM-8 surpasses those of both commercial water sorbents and promising MOFs (Supplementary Table 11), suggesting a unique opportunity to construct a compact and lightweight sorption system using this adsorbent under the premise of the well design of mass and heat transfer in the packed bed.

The production cost of water sorbents is also a key concern for the application of the ADC systems. The cost of reagent-grade raw materials for EMM-8 is one percent to one third of the outstanding MOFs considered here (Supplementary Table 12), indicating its extremely attractive advantage in commercialization potentials. Although the autoclave synthesis in fluoride-containing medium in this work may be a limitation of this material, methods of ionothermal synthesis, reuse of the organic templates can be employed to synthesize this sorbent in more cost-effective and environment-friendly way[50]. Accordingly, together with its outperforming energy efficiency, power density, and level of production maturity, the proposed EMM-8 clearly shows a successful combination of superior performance and economic potential.

## Discussion

This work demonstrates a zeotypic aluminophosphate EMM-8 as a high-performance water adsorbent that outperforms all the previously reported MOFs for ultra-low-temperature-driven refrigeration applications. EMM-8 exhibits excellent water-adsorption performances of high water uptake at an extremely narrow relative pressure range, fast sorption kinetics, facile regeneration, and remarkable stability. The abundant micropores with a uniform pore size of 0.7 nm and low water-adsorption enthalpy endow adsorption refrigeration systems with a notably high COP of 0.85 at a low driving temperature of 63 °C. Moreover, the adsorption-based system using EMM-8 shows a potential of high $SCP_{80\%}$ of $1.1\,kW\,kg_{sorbents}^{-1}$ under rigorous cooling conditions at the evaporation temperature as low as 5 °C and desorption temperature as low as 65 °C. The developed EMM-8 is expected to be a promising candidate for adsorption-based refrigeration, featuring excellent working performance and desirable stability, combined with unique advantages of scalable synthesis. This work paves a low-carbon way of developing a promising aluminophosphate adsorbent to realize energy-efficient ultra-low-temperature-driven refrigeration.

## Methods

**Typical synthesis of EMM-8**. The aluminum precursor, 0.662 g pseudo-boehmite, was slowly added to a mixture solution of 3.6 ml deionized water and 0.682 ml 85% phosphoric acid. The solution was strongly stirred at room temperature for 10 min to form a homogeneous suspension. Then, 0.112 ml of mineralizer 40% HF and 1.234 g 4-DMAPy were added to the mixture. After another 10 min of stirring and 10 min of ultrasonication, the gel was transferred to a 50 ml Teflon-lined stainless-steel autoclave. The reactor was heated and kept at 175 °C for 72–84 h under static

conditions. After cooling down to room temperature, the white powder of EMM-8 was collected by centrifugation, washed with deionized water, and dried in a vacuum oven. The synthesized samples were activated by calcination at 600 °C for 6 h with heating rates of 5 °C min$^{-1}$ to remove the SDA. Three samples obtained by repetitive fabrication progresses were used to demonstrate the consistency of the synthesis, as shown in Supplementary Fig. 20. After water adsorption, this AlPOs is regenerated at temperature above 65 °C.

**Characterization.** The powder XRD patterns were recorded at room temperature under ambient conditions with a BRUKER instrument (D8 Focus, Cu K$_\alpha$ with **k** = 1.5418 Å). The morphologies of samples were characterized by SEM (Hitachi S4800) and higher resolution transmission electron microscopy (JEM-2100F) equipped with an Energy dispersive X-ray detector. The surface morphology and thickness of the sample were analyzed by AFM (Bruker Multimode 8) using dynamic mode scanning at scan area $2 \times 2$ μm$^2$. Specific surface area and pore volume of samples were obtained by nitrogen gas adsorption at a low temperature of about 77 K using a gas adsorption analyzer (Quantachrome Quadrasorb SI-MP). Solid-state NMR spectra were recorded with an Advance III HD Bruker 500 NMR spectrometer (**B**$_0$ = 11.7 T, which corresponds to Larmor frequencies of 500.1, 130, and 202 MHz for $^1$H, $^{27}$Al, and $^{31}$P, respectively). The samples were packed either in 4 mm (for $^{27}$Al and $^{31}$P, MAS 10 and 13.3 kHz) and 2.5 mm (for $^1$H, MAS 30 kHz) outer diameter rotors. The surface composition was measured by XPS (ESCALAB 250Xi). Inductively coupled plasma optical emission spectroscopy analysis was carried out on a Thermo (iCAP Q) spectrometer. The structural stability of the samples was characterized by a thermogravimetric analyzer (TGA, TG-DTA6300). The thermal diffusivities (*a*) of dry sorbent tablets were measured by the laser flash method using a commercial instrument (LFA 447, Netzsch, GER). The thermal capacity (*C$_p$*) of dry composite sorbent was measured by using differential scanning calorimetry (Pyris1 DSC, Perkin-Elmer, Inc., USA). Then, the thermal conductivities of these tablets were obtained by equation $\lambda_{sorbent} = a \cdot \rho \cdot C_p$.

**Water-sorption measurement.** Water-sorption isotherms were measured by a 3H-2000 PW intelligent gravimetric analyzer (IGA, Beishide Instrument Technology Co., Ltd.) with the mass inaccuracy is ±0.001 mg. The accuracies of pressure sensor (INFICON, CDG025D) and temperature sensor (OMEGA PT100) are ±0.1 Pa and ±0.1 °C, respectively. The IGA was automatically operated to precisely control the water vapor pressure (1–95% RH) and temperature (20–60 °C). Prior to adsorption experiments, samples were completely dehydrated at 150 °C under high vacuum to a constant weight. The repeatability and uncertainty of the tests is confirmed by three measurements for one sample under the same conditions, as shown in Supplementary Fig. 21.

**Cyclic water adsorption/desorption tests.** Multiple cycles of water adsorption-desorption were also performed using a 3H-2000 PW intelligent gravimetric analyzer. Hydration/dehydration cycling tests were was carried out with the sequential procedure of isothermal measurements at 30 °C and relative humidity of 0.3 until reaching constant weight, followed by drying at 110 °C to a constant weight under high vacuum, which was repeated 30 times. In prior to the multiple cycle experiment, the first cycle was carried out by a different condition: EMM-8 is dehydrated at 150 °C overnight under vacuum, hydrated at 30 °C and 30% RH, and then dehydrated again at 110 °C under high vacuum.

**Kinetic measurements.** The water-sorption kinetics of sorbents were measured by using a self-constructed testing system based on volume method proposed by Aristov[50,51] (Supplementary Fig. 17). The absolute accuracies of pressure sensor (UNIK 5000 with accuracy of 0.04% over measurement range of 0–7 kPa, provided by Druck, GE company) and temperature sensor (OMEGA PT100) were ±2.8 Pa and ±0.1 °C, respectively. The EMM-8 grains were obtained by powder pressing without binder, manual grinding, and sieving, successively. The shaped EMM-8 granules with different sizes of 0.3–0.45, 0.45–0.6, and 0.8–0.9 mm (Supplementary Figs. 18, 19 and Table 6) were used in this work. The characteristic sorption time (τ), as well as SCP$_{max}$ and SCP$_{80\%}$ were calculated by the kinetic measurement data (Fig. 5f) at different boundary conditions for ADC applications (Supplementary Table 7), and more details can be found in Supplementary Note 3.

**Computational details.** The framework structure was constructed from their corresponding experimental single-crystal diffraction data[40]. First-principles calculations were first performed within the framework of density functional theory and the plane-wave pseudopotential functional approach[52,53], as implemented in the Vienna ab initio Simulation Package[54,55]. Ion-electron interactions were implemented with the projector-augmented wave method[56]. The generalized gradient approximation and the PBE functional[57] were used with a 450 eV limit for the plane-waves. All the atom positions were relaxed until the energy and force changes on each atom were less than 0.001 meV and 0.01 eV Å$^{-1}$, respectively. For geometry optimization and electronic structure calculations, the Brillouin zone was sampled with $2 \times 4 \times 4$ Γ-centered Monkhorst-Pack K-point grid. Based on the self-consistent charge and potential, the Density Derived Electrostatic and Chemical net atomic charges are calculated (Supplementary Table 13) using chargemol code[58]. GCMC simulations were employed by Sorption code in commercial software of Material Studio to calculate the water adsorption in the AlPO at 303 K. The material was treated as rigid frameworks with atoms frozen at their crystallographic positions, and the simulation box was made of 12 conventional unit cells ($2 \times 2 \times 3$). A cutoff radius was set to 1.2 nm for the LJ interactions, while the long-range electrostatic interactions were handled by the Ewald summation technique. The augmented CVFF force field parameter[59] was adopted to describe the LJ parameters for the atoms in the AlPOs framework, whereas water molecules were described by the simple SPC potential model[60] (Supplementary Table 14). This combination has been proven to be effective for the calculation of water-sorption isotherm in zeolite[61,62]. Periodic boundary conditions were considered in all three dimensions. For each state point, GCMC simulations consisted of $2 \times 10^8$ steps to ensure the equilibration, followed by $3 \times 10^8$ steps to sample the desired thermodynamic properties. In addition, to obtain accurate ensemble averages in GCMC simulations, at least millions of configurations generated by random translation, rotation, regrowth, and swap moves are sampled in each simulation.

## Data availability
Source data are provided with this paper.

## Code availability
The codes and simulation files that support the plots and data analysis within this paper are available from the corresponding author upon request.

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

## Acknowledgements

This work was supported by the National Natural Science Foundation of China (No. 51836009) and the National Key R&D Program of China (No. 2018YFE0100300).

## Author contributions

Z.L. contributed to the synthesis, shaping and general characterization of EMM-8, collected water-sorption data, calculated the thermodynamic efficiency of EMM-8 for water-sorption, evaluated specific cooling power values, and contributed to the writing of the paper. J.X. carried out the water-adsorption kinetics tests of EMM-8, helped with the manuscript preparation and revision. M.X. designed the study on synthesis and characterization, performed the GCMC simulation of the water-adsorption isotherms, analyzed all the results, contributed to the writing of the paper, and guided the structure of the full manuscript. C.H. contributed to the general characterization and collection of EMM-8. R.W. contributed to funding acquisition and supervised the evaluation of water-based sorption heating and cooling. T.L. contributed to funding acquisition, supervised, and directed the writing of the whole manuscript. X.H. contributed to funding acquisition and supervised the synthesis and general characterization of EMM-8. All the authors discussed the results and commented on the manuscript.

## Competing interests

The authors declare no competing interests.
