## [Peer Review File · Nature Communications]

REVIEWER COMMENTS

Reviewer #1 (Remarks to the Author):

This paper proposes a low-cost zeolite-like aluminophosphate with SFO 24 topology (EMM-8) for potential water-sorption-driven refrigeration. An extensive review of the literature on the subject matter was conducted and challenges were identified. However, what are the uniqueness, novelty and potential contributions to knowledge need to be judiciously documented in the introduction section? A comprehensive table is suggested for this purpose. A few points that the authors need to address include

1. It is highly inappropriate to present key results on EMM-8 such as COP values in the introduction section without any substantiation from any results. Please remove.
2. Perhaps one of the commercially adopted water-adsorbent materials is silica-gel. The authors need to explain their choice of benchmarking materials and the rationale for not selecting silica-gel.
3. The repeatability of the experimental results needs to be presented and documented.
4. Explain the reason for observable hysteresis in the sorption/desorption process of EMM-8.
5. Uncertainty analyses for all experimental data need to be conducted and error bars need to be displayed for all data points.
6. How does the porosity (pore density) of EMM-8 compare to other benchmarked materials such as MOF, zeolite, etc? It is a critical characteristics that need to be considered during water uptake and release.
7. Do provide other key other key properties including thermal conductivity, moisture diffusivity, specific heat capacities, apparent density etc.
8. The types of sensor used and their respective accuracies need to be described under supplementary Figure 16.
9. What are the typical COP values for the other benchmarking materials and how do they compared to EMM-8?
10. Provide some information on the activation energy of EMM-3 compared to the benchmarked adsorbent materials.
11. It is known that the performance of many heat-driven refrigeration/heat pump is affected by its cooling/heating capacity. Explain how EMM-8 would perform when incorporated to deliver a refrigeration or heat pump at different varying scale capacities.
12. Describe the key limitations of this study and how they are expected to be addressed.

Reviewer #2 (Remarks to the Author):

The manuscript reports a refrigeration system based on water adsorption, which is built using novel porous materials fabricated by the authors. These materials, zeolite-like porous aluminophosphates, provide high water uptake and help reaching out COP of 0.85, which is quite high for adsorption refrigerators (not sure if it a record though). This result looks very promising for potential application and is worth reporting in a high-profile journal. The paper is nicely written, and I read it with a great interest. Much of the details of the experimental procedures are provided, which would be probably sufficient to reproduce the results (not so much for the computational part). The conclusions are mostly supported by the data, etc. Overall, I would say the paper looks like a good fit to Nature Communications. However, I believe that several things should be taken care of before recommending the manuscript for publication. I am especially concerned with the comment #1 below. The rest, while being important, are just technical.

Major:

1. Please include the details on the regeneration procedure. In fact, use of HF for the regeneration is a concern, which might limit the practical use of the proposed materials. This issue should be discussed in detail.
2. Please include a table with all the values needed to calculate the COP and all the equations which are needed for it, so that one can easily follow
3. Details of GCMC simulations: what code was used? what water model was used? what EOS was used to relate chemical potential to pressures?
4. Computational data: all the DFT and GCMC data (input files and equilibrated configurations) should be documented as supporting data to secure the reproducibility

Minor:

SI P4: indexes are missing for

SI Fig 1(a,b): hard to read because of low quality/resolution

SI Fig 7(d): scale is not given

SI Fig 4, 13: T units are missing [same in other figures]

SI Fig 16: hard to read because of low quality/resolution

Reviewer #3 (Remarks to the Author):

Report on the paper »Record performance of ultralow-temperature-driven water-based sorption refrigeration enabled by low-cost zeolite-like porous aluminophosphate« by Liu et al.

The authors report on a performance of porous aluminophosphate with SFO topology in water-sorption driven heat transformation (refrigeration). The manuscript is very well written and the study of the material well performed. I am very glad that the authors recognized the potential of aluminophosphates in heat transformation applications and also tried to address one of their crucial drawbacks, which is the cost of materials (due to the use of mostly costly structure directing agents). At the same time they emphasized the structure stability of aluminophosphates in humid conditions and a nice attempt to scale up the synthesis.

However, there are some points which need to be addressed, before the manuscript is considered for publication in Nature Communications.

1. The authors compare ALPO-SFO with MOF materials and at some points also with SAPO-34 (aluminophosphate doped with Si). I think that more detailed comment on/comparison with pure aluminophosphates, like ALPO-TRIC (Ristic et al, *Advanced Functional Materials*, 2012, not cited in the manuscript) and ALPO-LTA (Krajnc et al., 2017, cited in the Introduction and mentioned in Figure 5, Supp Fig. 15), would be appropriate. It is true that both listed materials were mostly evaluated for water-sorption-based heating applications (ALPO-LTA was evaluated for cooling in conference contribution Eurosun, 2018, Ristic et al.), but the basic principles are the same. Furthermore, they are all pure aluminophosphates and not Si-doped version (doping can crucially change sorption characteristics in low pressure range). Additionally, the ALPO-SFO, ALPO-TRIC and ALPO-LTA were all prepared in F-media (which adds to relaxation of the SBU) and it is anticipated that this also contributes to final water sorption mechanism of activated material. How do the authors explain the selection of materials to compare with and how they explain the obtained sorption mechanism (S-shape isotherm) of their material? I am quite sure, it is not only pore opening size.

2. S-shape isotherm is mentioned as optimal type for selected applications. This “requirement” is many times misinterpreted in the literature. It is considered beneficial, because it indicates that material would be dehydrated at lower T. However, we could conclude it from de-sorption part. Sometimes, there is a strong hysteresis between adsorption and desorption curves. Could authors provide de-sorption isotherm?
3. Structure stability in humid conditions; do authors consider 24h test at 100 deg.C appropriate/long enough? Cycling stability test, which is relevant for application – it is mentioned in the manuscript, but more information on the procedure are needed? Comment – thermal stability up to 700 deg. C is compared with MOFs, but ALPO-TRIC is thermally stable up to 900 deg. C.
4. Price of material (Supp. Table 9); Did authors consider the cost of autoclave synthesis vs. autoclave-free synthesis (MOFs mostly). Can the authors comment the IL synthesis of ALPOs, which is also considered greener?
5. I guess the 0.36 g·g⁻¹ water uptake is maximal water uptake at p/p₀=0.8. What is the uptake at the S-step (applicable for application)?
6. Page 4: there is at least one more aluminophosphate developed, i.e. AQSOA-Z05 at the beginning (in 2005).
7. Page 5, end of first paragraph: Could authors just shortly explain which types and characteristics of MOFs, that are better than some already mentioned aluminophosphates, for example ALPO-LTA.
8. Supp. Table 3: Please check the data for ALPO-LTA.

Response to the Reviewer' Comments

Reviewer 1: This paper proposes a low-cost zeolite-like aluminophosphate with SFO topology (EMM-8) for potential water-sorption-driven refrigeration. An extensive review of the literature on the subject matter was conducted and challenges were identified. However, what are the uniqueness, novelty and potential contributions to knowledge need to be judiciously documented in the introduction section? A comprehensive table is suggested for this purpose. A few points that the authors need to address include

Reply: First of all, we would like to thank the reviewer for giving positive comments on our work and for your carefully evaluation on the manuscript. We appreciate your detailed review to help us to improve the manuscript. Please find the point-to-point answers to your comments in the following.

Water-sorption-driven refrigeration has attracted tremendous attentions as a near-zero carbon emission cooling strategy to address the urgent global challenge of climate change. Its widespread applications require low-cost high-performance porous materials. However, traditional sorbents (e.g., silica gels and zeolites) show low water adsorption capacities and high regeneration temperatures, resulting in poor energy efficiency for adsorption systems. In recent decade, several metal-organic frameworks (MOFs) were reported as new effective water adsorbents for both heating and cooling applications. Although the reported results show good feasibility of MOFs, their raw materials are usually expensive and their scaled production capacity is questionable, indicating the difficulty of the widespread application of MOFs at present. Aluminophosphates (AIPOs) are regarded as promising alternatives to realize high-efficient low-cost adsorption cooling. The lab-scale and commercial-scale adsorption chiller using AIPOs (e.g. SAPO-34) show superiorities in both technical and economic potentials; however, their low working capacity ($<0.3 \text{ ml}\cdot\text{ml}^{-1}$) and comparable high driven temperature ($>80 \text{ }^\circ\text{C}$) limit the energy efficiency and power density of the chillers. Therefore, to overcome the above drawbacks, we developed a novel SFO-type AIPOs, EMM-8, in this work.

As your suggestion, we emphasized the novelty and contributions of this work in the introduction section, summarized as follows: i) We report the first-ever use of a lesser-known SFO-type AIPO, namely EMM-8, as an efficient water adsorbent for ultra-low-temperature-driven adsorption refrigeration. Water adsorption evaluations highlight that this sorbent exhibits very steep water uptake, large working capacity ($\sim 0.4 \text{ ml}\cdot\text{ml}^{-1}$), facile regeneration, fast adsorption/desorption kinetics, and remarkable hydrothermal stability. To the best of our knowledge, EMM-8 shows the highest coefficient of performance (COP) reaching 0.85 for cooling and 1.75 for heat pump among the reported porous materials. Importantly, the driving temperature for EMM-8 ($63 \text{ }^\circ\text{C}$, cooling application from $30 \text{ }^\circ\text{C}$ to $5 \text{ }^\circ\text{C}$) is $5\text{--}15 \text{ }^\circ\text{C}$ lower than those of the state-of-the-art materials, making EMM-8 more suitable for the utilization of low-grade thermal energy; ii) We demonstrated a 100 g-scale synthesis of EMM-8 using the cheap raw materials (only one percent of the conventional MOFs) without performance degradation, suggesting its low-cost and scalable fabrication. This work provides a more-effective low-cost water adsorbent realizing high-performance cooling at ultralow driven temperatures; and iii) Combining GCMC simulation and experiment, we clearly show that the exceptionally high energy performance originates from a considerably low isosteric heat of adsorption contributed

by the hydrogen-bonded water clusters in the pores of EMM-8. The water adsorption mechanisms and the structure-property relationship of the sorbent are discussed in this paper, which will guide researchers in relative fields to design new more-effective water sorbents.

Supplementary Table 4, Table 5, Table 11, and Table 12 listed the detailed comparison of the proposed EMM-8 with state-of-the-art sorbents in the perspectives of working performance and costs.

1. It is highly inappropriate to present key results on EMM-8 such as COP values in the introduction section without any substantiation from any results. Please remove.

Answer: Thanks for your constructive suggestion. We agree with your comments and have removed the theoretically calculated values, including COP values, of EMM-8 in the introduction section of the revised manuscript.

Changes add to manuscript:

P6: Herein, we report the first-ever use of EMM-8 as an efficient water adsorbent for ultra-low-temperature-driven heating and cooling. We thoroughly characterize it towards water adsorption application and achieve a 100 g-scale fabrication. By comprehensive water sorption evaluation, we find the EMM-8 represents step-wise water adsorption facile desorption at low temperature, and desirable hydrothermal stability. More importantly, EMM-8 shows a notably high COP for cooling and heating at lower driven temperatures than those of the best current state-of-the-art materials, indicating a promising application potential for the exploitation of yet mostly unused low-grade heat. To elaborate the mechanism of high efficiencies at ultralow temperatures, we carried out the theoretical simulation together with experimental measurements, uncovering the high COP originates from a considerably low isosteric heat of adsorption contributed by the hydrogen-bonded water clusters within the pores of EMM-8. This work provides a more effective water adsorbent for realizing affordable, scalable, and high-performance adsorption heating and cooling.

2. Perhaps one of the commercially adopted water-adsorbent materials is silica-gel. The authors need to explain their choice of benchmarking materials and the rationale for not selecting silica-gel.

Answer: Thanks for your valuable comments. Silica-gel, as a commercially available material for adsorption heating and cooling systems, has been extensively investigated in the previous studies. It shows advantages of acceptable price and desirable stabilities, but obvious drawbacks are identified, including low water uptake and insufficient working capacities, resulting in poor energy efficiency and low power density for adsorption systems. The exiting prototypes of adsorption chillers/heat pumps with silica gel-water pair show low COP_c of $\sim 0.3-0.6$ and COP_h of $\sim 1.1-1.5$. Also, the SCP/SHP of these devices are relatively poor ($< 0.4 \text{ kW}\cdot\text{kg}^{-1}$), resulting in the bulky volume and heavy weight for a full-scale equipment. Therefore, the application of the silica gel-water adsorption chiller/heat pump is limited.

Developing the more-effective low-cost water sorbents to achieve the high-efficient adsorption heating and cooling have triggered a new round of worldwide research. During the

last decades, metal-organic frameworks (MOFs) with highly tunable pore structures and hydrophilic/hydrophobic functional groups, are considered as new effective water adsorbents. Lots of new MOFs with high water adsorption capacities have been reported. Many of them, like MOF-801, MOF-303, MIP-200, MIL-160, CAU-23, Zr-fum, Co-CUK-1, and KMF-1, show notably higher energy efficiency than silica gel. However, challenges from expensive raw materials and low-scalability in synthesis make MOFs unaffordable in this stage. Hence, in this work, a low-cost and scalable water sorbent is proposed, which exhibit the record COP_c of 0.85 for adsorption cooling. The best-in-class benchmarking materials were selected to reveal the record performance of the proposed material. Considering the apparent low working performance of silica gel; thus, it was not selected for comparison in our original manuscript. Now, silica-gel was added as a benchmarking material in the revised manuscript for more rational and comprehensive comparison. Please see **Supplementary Figure 14 and Supplementary Table 5**.

Supplementary Figure 14. Coefficient of performance values for heating and cooling in comparison with reference materials. a. Refrigeration conditions used were $T_{ev} = 5\text{ }^{\circ}\text{C}$ and $T_{con} = 30\text{ }^{\circ}\text{C}$. **b.** Heat pump conditions used for calculations were $T_{ev} = 15\text{ }^{\circ}\text{C}$ and $T_{con} = 45\text{ }^{\circ}\text{C}$. EMM-8 (■), MOF-801 (◆)^{1,2}, MIP-200 (▲)³, MIL-160 (▼)⁴, CAU-10 (◇)^{1,5}, Co-CUK-1 (♣)⁶, KMF-1 (★)⁷, Zr-Fum HT (●)⁸, AIPO-LTA (□)⁹, FAPO-5 (▷)¹, SAPO-34 (◁)¹, silica gel (*)¹⁰.

Supplementary Table 5. Water sorption properties and specific energy capacity of EMM-8 and reference water adsorbents.

Material	Crystal density (g·cm ⁻³)	V ^a (cm ³ ·g ⁻¹)		Porosity	<-Δ _{ads} H> (kJ·mol ⁻¹)	Working capacity ^{d,f}		Specific energy capacity ^{e,f}		Ref
		V _{mic}	V _{total}			(g·g ⁻¹)	(ml·ml ⁻¹)	(Wh·kg ⁻¹)	(kWh·m ⁻³)	
EMM-8	1.50	0.30	0.59	0.45	46.76	0.256	0.384	176	263	This work
MOF-801	1.59	0.45	/	0.72	58.40	0.056	0.089	38	61	1,2
MIP-200	1.16	0.40 ^b	/	0.46	51.20 ^f	0.187	0.217	128	148	3
MIL-160	1.07	0.398	/	0.43	50.64 ^f	0.066	0.071	45	48	4
CAU-10	1.15	0.27	/	0.31	53.50	0.293	0.337	201	231	1, 5
CAU-23	1.07	0.48	/	0.51	48.20	0.262	0.280	180	193	11
Co-CUK-1	1.46	0.26 ^c	/	0.38	48.65 ^f	0.045	0.066	31	45	6
Co ₂ Cl ₂ -(BTDD)	0.69	/	/	/	46.43 ^f	0.119	0.082	81	56	12
COF-TpPa	/	/	/	/	45.00	0.215	/	147	/	13
KMF-1	1.08	/	0.473	/	52~57	0.051	0.055	35	38	7
Zr-Fum HT	1.59	0.389	0.553	0.62	45~52	0.051	0.081	35	56	8
AIPO-LTA	1.41	0.32	/	0.45	55.58 ^f	0.124	0.175	85	120	9, 14
FAPO-5	1.75	0.15	/	0.26	56.10	0.165	0.289	113	198	1, 15
SAPO-34	1.43	0.212	/	0.30	57.00	0.038	0.054	26	37	1, 16
AIPO-5	1.75	/	/	/	52.6	0.135	0.236	92	162	1
Silica gel	0.72	0.38	0.40	0.27	54.64 ^f	0.076	0.055	52	37	10

^a Based on N₂ sorption isotherms at 77 K.

^b After a simple washing in boiling water or in aqua regia at room temperature.

^c Based on CO₂ sorption isotherms at 194.5 K.

^d Working capacity deduced from one refrigeration cycle at T_{ev}= 10 °C, T_{con}= 30 °C, and T_{des}=65 °C;

^e Heat transferred from the evaporator in one refrigeration cycle at T_{ev}= 10 °C, T_{con}= 30 °C and T_{des}= 65 °C;

^f Values for MOF-801, MIP-200, MIL-160, CAU-10, CAU-23, Co-CUK-1, Ni-CUK-1, Mg-CUK-1, Co₂Cl₂-(BTDD), COF-TpPa, KMF-1, Zr-Fum HT, AIPO-LTA, FAPO-5, SAPO-34, AIPO-5, Silica gel were calculated by characteristic curves and water adsorption data taken from references.

Changes add to Supplementary Information:

P24 and P39: Silica-gel has been added as a benchmarking material to **Supplementary Figure 14** and **Supplementary Table 5**.

3. The repeatability of the experimental results needs to be presented and documented.

Answer: Thanks for your valuable comments. We have added the demonstration about the repeatability of the experimental results from two aspects. First, we present the repeatability of the synthesis and characterizations of this proposed material. Three samples obtained by repetitive fabrication progresses were used to show the consistency of the synthesis. The XRD, BET, and SEM results indicate that these three samples exhibit the similar crystallinity, texture property, and morphologic characterization, as shown in the following Figure R1. A robust 100

g-scale fabrication of this aluminophosphate is also achieved (**Supplementary Figure 7b**), further suggesting the well repeatability of the synthesis. Hence, it is clear that the synthesis of the material is easy to be reproduced. Second, we present the repeatability of testing results of water adsorption isotherms and kinetics, which are the keys to evaluate the COP and SCP. We performed three tests of water adsorption isotherms and kinetics by using the same samples, methods, and equipment. As shown in the following Figure R2, the results indicate the nearly identical water sorption performance with small measurement errors (Please see errors analysis in **Answer 5**), including the water adsorption capacity, the transition relative pressure of water uptake in isotherms, and water adsorption/desorption rate.

We have added these results to demonstrate the repeatability of the experimental results in the revised Supplementary Information (**Supplementary Figure. 20**).

Figure R1. Preparation repeatability. **a.** XRD patterns of samples 1-3 obtained by repetitive fabrication progress three times. **b.** Nitrogen sorption isotherms of samples 1-3 taken at 77 K, the inset image gives the pore size distribution of the samples. **c.** SEM images of samples 1-3.

Figure R2. Repeatability of water adsorption test. a. Water adsorption isotherm of three tests at 30 °C. b. Dimensionless water uptake curves during adsorption (30 °C and 870 Pa) and desorption (80 °C and 4200 Pa) of three tests. The size of EMM-8 grain is 0.45~0.6 mm.

Changes add to manuscript:

P20: Three samples obtained by repetitive fabrication progresses were used to demonstrate the consistency of the synthesis, as shown in **Supplementary Fig. 20**.

Changes add to Supplementary Information:

P30-31: **Fig. R1** was added to **Supplementary Information** as **Supplementary Figure 20**. **Fig. R2** was added to **Supplementary Information** as **Supplementary Figure 21a**.

4. Explain the reason for observable hysteresis in the sorption/desorption process of EMM-8.

Answer: Thanks for your valuable comments. A small hysteresis is observed in the desorption branch of EMM-8 in the narrow range of P/P₀, which is attributed to the presence of the mesoporous. Based on nitrogen adsorption results, EMM-8 exhibits a large mesopores volume of as high as 0.29 cm³·g⁻¹, which is close to the micropore volume. Capillary condensation in the mesopores leads to the occurrence of the desorption hysteresis. SEM and AFM observations (**Fig. 2b and f-h**) reveal that the thicknesses of the EMM-8 nanoplates are distributed between 20 nm and 120 nm. We have added some explanation about the desorption hysteresis phenomenon in the revised manuscript.

Figure R3. Adsorption/desorption isotherms for EMM-8 at 30 °C

Changes add to manuscript:

P11: A small hysteresis is observed in the desorption branch in the narrow range of $P/P_0=0.15-0.2$ (Supplementary Fig. 9c), which may be attributed to the presence of the mesoporous.

Changes add to Supplementary Information:

P19: Fig. R3 was added to Supplementary Information as Supplementary Figure 9c

5. Uncertainty analyses for all experimental data need to be conducted and error bars need to be displayed for all data points.

Answer: Thanks for your valuable comments. As your suggestion, we conducted the uncertainty analysis based on the standard error analysis method. Water sorption isotherms were measured by a 3H-2000 PW intelligent gravimetric analyzer (IGA, Beishide Instrument Technology Co., Ltd.). The electronic balance of intelligent gravimetric analyzer is customized, and its mass inaccuracy is ± 0.001 mg. The inaccuracies of pressure sensor (INFICON, CDG025D) and temperature sensor (OMEGA PT100) are ± 0.1 Pa and ± 0.1 °C, respectively. A confidence level of the all uncertainties is 95%. Based on Equation R(1)-R(6), the uncertainty analysis of water uptake q ($\text{g}\cdot\text{g}^{-1}$) at 30 °C was carried out, as shown in Fig. R1, where m , m_0 , q_i , and \bar{q} represent the real-time quality of EMM-8 with the adsorption process, the quality of totally dehydrated EMM-8, the water uptake for multiple measurement, and the average water uptake of multiple measurement, respectively. μ and E represent the uncertainty and relative uncertainty. As shown in Fig. R4, the uncertainty of water uptake is in the range of 0.22~1.80 % at relative pressure $P/P_0 \leq 0.16$, which corresponds to the main region of interest. For example, the uncertainties of water uptake q ($\text{g}\cdot\text{g}^{-1}$) at $P/P_0 = 0.02$ and $P/P_0 = 0.16$ are 1.80 % and 0.22 %, respectively. However, this value drops to less than 0.1% at higher relative pressure $P/P_0 > 0.16$. Accordingly, we think the testing error is small enough to be acceptable. As your suggestion, the error bars were given in Fig. R5.

$$\mu_{q,A} = t \sqrt{\frac{\sum_{i=1}^n (q_i(P,T) - \bar{q})^2}{n(n-1)}} \quad \text{R(1)}$$

$$q(P, T) = \frac{m - m_0}{m_0} \quad \text{R(2)}$$

$$\ln q(P, T) = \ln(m - m_0) - \ln m_0 \quad \text{R(3)}$$

$$E_{q,B} = \sqrt{\left(\frac{\partial \ln q(P, T)}{\partial m_0} \mu_{m_0}\right)^2 + \left(\frac{\partial \ln q(P, T)}{\partial m} \mu_m\right)^2} = \sqrt{\left(-\frac{m \mu_{m_0}}{m_0(m - m_0)}\right)^2 + \left(\frac{\mu_m}{m - m_0}\right)^2} \quad \text{R(4)}$$

$$\mu_{q,B} = E_{q,B} \times q(P, T) \quad \text{R(5)}$$

$$\mu_q = \sqrt{\mu_{q,A}^2 + \mu_{q,B}^2} \quad \text{R(6)}$$

Figure. R4. Uncertainties of water uptake q for EMM-8 at 30 °C in relation to P/P_0

Fig. R5 shows the water adsorption capacity with error bars at 30 °C for three parallel experiments (the water adsorption isotherms of three parallel tests are shown in the Fig. R2 in Answer 3).

Figure. R5. Water uptake q with error bars for EMM-8 at 30 °C in relation to P/P_0

Changes add to manuscript:

P22: The repeatability and uncertainty of the tests is confirmed by three measurements for one sample under the same conditions, as shown in **Supplementary Fig. 21**.

Changes add to Supplementary Information:

P31-32: **Fig. R5** were added to **Supplementary Information** as **Supplementary Figure 21b**. Uncertainty analysis was added to **Supplementary Information**.

6. How does the porosity (pore density) of EMM-8 compare to other benchmarked materials such as MOF, zeolite, etc? It is a critical characteristics that need to be considered during water uptake and release.

Answer: Thanks for your valuable comments. We have added the porosity and pore volume of EMM-8 and the benchmarking materials in the revised manuscript (Please see **Supplementary Table5**). EMM-8 has a high porosity of 0.45 and a micropore volume of $0.30 \text{ cm}^3 \cdot \text{g}^{-1}$, comparable to that of the other AlPOs and the best-in-class MOFs, including SAPO-34 ($0.30, 0.212 \text{ cm}^3 \cdot \text{g}^{-1}$), AlPO-LTA ($0.45, 0.32 \text{ cm}^3 \cdot \text{g}^{-1}$), Co-CUK-1 ($0.38, 0.26 \text{ cm}^3 \cdot \text{g}^{-1}$), and CAU-10 ($0.31, 0.27 \text{ cm}^3 \cdot \text{g}^{-1}$). In particular, EMM-8 exhibits the large mesopores volume of as high as $0.29 \text{ cm}^3 \cdot \text{g}^{-1}$, suppressing most of the benchmarking materials. It offers a high potential to achieve a faster transfer of water molecules within the materials, which has been confirmed by the kinetic measurements as discussed in the section about SCP. We have added some discussions about the comparison of the pore volume between this proposed material and other selected sorbents.

Changes add to manuscript:

P7: The EMM-8 shows high porosity of 0.45 and micropore volume of $0.3 \text{ cm}^3 \cdot \text{g}^{-1}$, comparable to that of the best current water adsorption sorbents (**Supplementary Table 5**).

Changes add to Supplementary Information:

P39: Porosity and Pore volumes of EMM-8 and the benchmarking materials were added to **Supplementary Table5**.

7. Do provide other key other key properties including thermal conductivity, moisture diffusivity, specific heat capacities, apparent density etc.

Answer: Thanks for your valuable comments. We have added the key properties mentioned by the reviewer in the revised manuscript. The apparent density of loose EMM-8 powder is about $0.49 \text{ g} \cdot \text{ml}^{-1}$. For thermal conductivity, we measured the thermal diffusivities (α) of dry sorbent tablet by the laser flash method using a commercial instrument (LFA 447, Netzsch, GER). The thermal capacity (C_p) of dry composite sorbent was measured by using differential scanning calorimetry (Pyris1 DSC, Perkin-Elmer, Inc., USA). Then, the thermal conductivities of these tablets are obtained by following equation.

$$\lambda_{\text{sorbent}} = \alpha \cdot \rho \cdot C_p$$

The EMM-8 pellet with packing density of 0.93 g ml^{-1} shows the thermal conductivities of $0.082 \sim 0.37 \text{ W} \cdot (\text{m} \cdot \text{K})^{-1}$ at $25 \sim 100 \text{ }^\circ\text{C}$ (**Fig. R6a**). These values are close to that of packed beds of commercial silica gel ($0.08 \sim 0.15 \text{ W} \cdot (\text{m} \cdot \text{K})^{-1}$) and the MOFs ($0.06 \sim 0.12 \text{ W} \cdot (\text{m} \cdot \text{K})^{-1}$). The dry EMM-8 represents the specific heat capacities of $0.63 \sim 0.93 \text{ J} \cdot (\text{g} \cdot \text{K})^{-1}$ (**Fig. R6b**), comparable to that of the

most of the porous materials, which is in the range from 0.6 to 1.1 J·(g·K)⁻¹.

Figure. R6. a. Thermal conductivity, and b. heat capacity of EMM-8 at 25~100 °C.

Base on the water adsorption kinetic curves, we calculated the diffusion coefficient of water vapor (moisture diffusivity) within the particles by using the following equation¹⁷:

$$q = \frac{\Delta W(t)}{\Delta W_{t \rightarrow \infty}} = \frac{12}{D} \sqrt{\frac{D_M t}{\pi}} \quad \text{R(7)}$$

where D_M (cm²·s⁻¹) is the effective intracrystalline diffusion coefficient, q is the dimensionless conversion, D (mm) is the particle diameter. Plot q versus $t^{1/2}$, and then a straight line with slope $\frac{12}{D} \sqrt{\frac{D_M t}{\pi}}$ can be obtained. Thus, diffusion coefficient D_M can be calculated by linear fitting the straight lines.

Fig. R7 plots water vapor conversion ratio q versus $t^{1/2}$ at 30 °C and 1200 Pa for different particle diameters (0.3~0.45 mm, 0.45~0.6 mm, and 0.8~0.9 mm). **Fig. R8** plots q for EMM-8 diameter 0.45~0.6 mm versus $t^{1/2}$ at 30 °C under different vapor pressures (870Pa, 1200Pa, and 1700Pa). The slopes of the lines are fitted, which was used to calculate the efficient diffusion coefficients D_M . **Table R1** shows the efficient diffusion coefficients D_M under different adsorption conditions and particle sizes.

Fig. R7. Water vapor conversion ratio q as a function of $t^{1/2}$ at 30 °C and 1200 Pa for different particle sizes (0.3~0.45 mm, 0.45~0.6 mm, and 0.8~0.9 mm).

Fig. R8. Water vapor conversion ratio q as a function of $t^{1/2}$ at 30 °C for 0.3~0.45 mm particle under different pressure (870Pa, 1200Pa, and 1700Pa).

Table R1. Diffusion coefficient ($D_M \sim 10^{-7} \text{ cm}^2 \cdot \text{s}^{-1}$) of water vapor on EMM-8

Cycle	Operating condition			Slope	$\frac{12}{D\sqrt{\pi}}$	D_M
	T (°C)	P (Pa)	Diameter D (mm)	($\text{s}^{-1/2}$)	(mm^{-1})	($10^{-7} \text{ cm}^2 \cdot \text{s}^{-1}$)
1	30	1200	0.3~0.45	0.0985	18.0528	2.98
2	30	1200	0.45~0.6	0.0787	12.8950	3.72
3	30	1200	0.8~0.9	0.0686	7.9646	7.42
4	30	870	0.45~0.6	0.0628	12.8950	2.37
5	30	1700	0.45~0.6	0.1154	12.8950	8.01

Changes add to manuscript:

P8: The thermal conductivity of EMM-8 was measured by the laser flash method at various temperatures. The EMM-8 pellets with packing density of 0.93 g ml^{-1} show the thermal conductivity of $0.082 \sim 0.37 \text{ W} \cdot (\text{m} \cdot \text{K})^{-1}$ at $25 \sim 100 \text{ °C}$ (**Supplementary Fig. 4a**). This value is close to that of adsorbent packed beds of commercial silica gel ($0.08 \sim 0.15 \text{ W} \cdot (\text{m} \cdot \text{K})^{-1}$) and the MOFs ($0.06 \sim 0.12 \text{ W} \cdot (\text{m} \cdot \text{K})^{-1}$). The specific heat capacity of dry EMM-8 was also measured via DSC method at room temperature. This material represents the specific heat capacity of $0.63 \sim 0.93 \text{ J} \cdot (\text{g} \cdot \text{K})^{-1}$ (**Supplementary Fig. 4b**), comparable to that of the most of the porous materials, which is in the range from 0.6 to $1.1 \text{ J} \cdot (\text{g} \cdot \text{K})^{-1}$.

P18: We calculated the diffusion coefficient of water vapor within the particles base on the water adsorption kinetic curves (**Supplementary Note 3**), as shown in **Supplementary Table 8**.

P21: The thermal diffusivity (a) of dry sorbent tablets was measured by laser flash method using a commercial instrument (LFA 447, Netzsch, GER). The thermal capacity (C_p) of dry composite sorbent was measured by using differential scanning calorimetry (Pyris1 DSC, Perkin-Elmer, Inc., USA). Then, the thermal conductivities of these tablets were obtained by equation

$$\lambda_{\text{sorbent}} = a \cdot \rho \cdot C_p.$$

Changes add to Supplementary Information:

P14: **Fig. R6** was added to **Supplementary Information** as **Supplementary Figure 4**.

P43: **Table. R1** was added to **Supplementary Information** as **Supplementary Table8**.

8. The types of sensor used and their respective accuracies need to be described under supplementary Figure 16.

Answer: Thanks for your useful comments. According to your suggestion, we have added the types of sensors used in adsorption kinetic measurements and their accuracies. The water adsorption-desorption kinetics of samples were measured by the self-constructed testing set-up based on volume methods using the high-accuracy pressure sensors of UNIK 5000 with accuracy of 0.04% over measurement range of 0-7 kPa, provided by Druck, GE company. The absolute accuracy of pressure sensor is $\pm 2.8 \text{ Pa}$. The accuracy of temperature sensor (OMEGA PT100) is

±0.1 °C

Changes add to manuscript:

P22: The water sorption kinetics of sorbents were measured by using a self-constructed testing system based on volume method proposed by Aristov (**Supplementary Fig. 17**). The absolute accuracies of pressure sensor (UNIK 5000 with accuracy of 0.04% over measurement range of 0-7 kPa, provided by Druck, GE company) and temperature sensor (OMEGA PT100) were ±2.8 Pa and ±0.1 °C, respectively.

9. What are the typical COP values for the other benchmarking materials and how do they compared to EMM-8?

Answer: Thanks for your comments. In this work, we compared the COP values of the EMM-8 and other benchmarking materials for cooling and heating, as shown in **Figure 5a, d** and **Supplementary Figure 14**. More detailed comparison was shown in the following **Table R2**, which was also added in the revised manuscript as **Supplementary Table 4**. The results indicate that EMM-8 exhibits an exceptionally high COP_c of 0.85 at the ultralow driving temperature of 63 °C, remarkably exceeding the state-of-the-art materials, including both commercially adopted silica-gel (0.52) and the recently reported best-in-class adsorbents, such as AIPO-LTA (0.75), MIP-200 (0.78), KFM-1 (0.75), and Co-CUK-1 (0.83). Moreover, under a standard heat pump condition, i.e., $T_{ev} = 15$ °C and $T_{con} = 45$ °C, EMM-8 also gives a very high COP_H of 1.75 at a driven temperature of 82 °C (**Fig. 5d** and **Supplementary Fig. 14b**). The comparison of this value between EMM-8 and other excellent materials confirms that EMM-8 outperforms both other AIPOs and most MOFs for heat pump applications. This value is as high as that of the best water adsorbents reported so far, i.e., Co-CUK-1 (1.77) and KMF-1 (1.74).

Table R2. Coefficient of performance of EMM-8 and reference water adsorbents.

Materials	COP for cooling ($T_{ev} = 5\text{ }^{\circ}\text{C}$, $T_{con} = 30\text{ }^{\circ}\text{C}$)			COP for heating ($T_{ev} = 15\text{ }^{\circ}\text{C}$, $T_{con} = 45\text{ }^{\circ}\text{C}$)			ref	
	COP _{c,max} (T_{desr} , $^{\circ}\text{C}$)	different T_{des} ($^{\circ}\text{C}$)			COP _{h,max} (T_{desr} , $^{\circ}\text{C}$)	different T_{des} ($^{\circ}\text{C}$)		
		65	70	80		80		100
EMM-8	0.85 (65)	0.85	0.84	0.82	1.75 (82)	1.65	1.72	This work
MOF-801	0.65 (86)	0.30	0.49	0.64	1.58 (100)	/	1.58	1,2
MIP-200	0.78 (70)	0.74	0.78	0.78	1.59 (96)	1.15	1.59	3
MIL-160	0.73 (97)	0.50	0.60	0.71	1.68 (108)	/	1.66	4
CAU-10	0.75 (71)	0.68	0.74	0.735	1.28 (102)	/	1.18	1,5
CAU-23	0.41 (82)	0.32	0.37	0.40	1.30 (117)	1.04	1.28	11
Co-CUK-1	0.83 (71)	0.57	0.83	0.81	1.77 (87)	1.11	1.76	6
Co ₂ Cl ₂ -(BTDD)	0.75 (97)	0.54	0.61	0.70	1.64 (122)	1.14	1.59	12
COF-TpPa	0.73 (72.5)	0.72	0.73	0.72	1.55 (104)	1.11	1.54	13
KMF-1	0.75 (70)	0.42	0.75	0.74	1.74 (85)	1.02	1.73	7
Zr-Fum HT	0.83 (80)	0.48	0.73	0.83	1.76 (93.5)	1.09	1.76	8
AIPO-LTA	0.76 (70)	0.69	0.76	0.75	1.72 (87)	1.03	1.71	9
AQSOA-Z01	0.70 (62)	0.69	0.685	0.66	1.36 (88)	1.20	1.33	1
AQSOA-Z02	0.68 (86)	0.48	0.62	0.68	1.61 (105~107)	1.41	1.61	1
AQSOA-Z05	0.11 (89)	0.086	0.109	0.109	1.01 (108)	/	1.005	1
Silica gel	0.52 (77)	0.45	0.50	0.51	1.41 (96)	1.11	1.41	10

Changes add to Supplementary Information:

P38: Table. R2 was added to **Supplementary Information** as **Supplementary Table4**.

10. Provide some information on the activation energy of EMM-8 compared to the benchmarked adsorbent materials.

Answer: Thanks for your meaningful comments. To the best knowledge of us, the activation energy based on the kinetics of desorption process is rarely reported for the best current water sorbents. Hence, the comparison of the activation energy of EMM-8 and the benchmarking adsorbent materials is difficult to carry out. In this work, the water sorption kinetics of EMM-8 were measured, and the SCP values were evaluated and compared with the benchmarking sorbents, as shown in Supplementary Table 11. The SCP value of EMM-8 surpasses those of both commercial water sorbents and promising MOFs, demonstrating the huge potential for water-based sorption cooling application. We would like to carry out the comparison of activation energy between EMM-8 and other sorbents in another work to further evaluate the kinetics performance of this proposed material.

For a water-based sorption chiller, a reversible water adsorption/desorption process is used to deliver the low-temperature waste heat or solar energy to cooling. Therefore, for a specific sorbent, the heat of adsorption plays an important role in its cooling performance. In this present work, we calculated the average adsorption enthalpy of EMM-8 through the water adsorption isotherms at varying temperatures, and compared its value with the benchmarked adsorbent materials, as shown in Supplementary Table 5. The results indicated that EMM-8 showed the lower adsorption enthalpy, suggesting the superior cooling performance due to the fact that the

theoretical maximum COP value equals with the ratio of water evaporation enthalpy and adsorption enthalpy. The water adsorption mechanism to explain the low adsorption enthalpy is also revealed by GCMC simulations. The formation of the H-bond water clusters and the appropriately large pore opening size of EMM-8 contribute to the low adsorption enthalpy and thus notably high COP for cooling. Please see the detailed description in the second paragraph in P11 of the revised manuscript.

11. It is known that the performance of many heat-driven refrigeration/heat pump is affected by its cooling/heating capacity. Explain how EMM-8 would perform when incorporated to deliver a refrigeration or heat pump at different varying scale capacities.

Answer: Thanks for your meaningful comments. The cooling/heating capacity of the thermally-driven adsorption refrigeration/heat pump is significantly influenced by the scale of the device. The mass and heat transfer characteristics, which is closely related to the size of the materials and equipment, play the important role in the performance of the water-based sorption refrigeration/heat pump, especially for the specific cooling/heating power (SCP or SHP). Radu et al.¹⁸(*Int. J. Heat Mass Transf.* 105, 326-337 (2017)). and Ammann et al.¹⁹(*Int. J. Heat Mass Transf.* 130, 25-32 (2019)) investigated the effect of heat and/or mass transport limitations on the power density of the adsorption heat pump. The results indicate that effective diffusivity within the material, thermal conductivity, and packing configurations are the critical factors for determining the specific cooling power of the adsorbent. In this work, the SCP of the proposed sorbent was evaluated by water sorption kinetic measurements of shaped granules with size of 0.45-0.6 mm, which is widely used to evaluate the cooling capacity on the material scale. The results indicate that a superior high SCP_{max} of 2.22 kW·kg_{sorbents}⁻¹ and SCP_{80%} of 1.1 kW·kg_{sorbents}⁻¹ for this material under the rather harsh working conditions ($T_{ev}=5$ °C, $T_{con}=30$ °C, and $T_{des}=65$ °C), surpassing those of both commercial water sorbents and promising MOFs. In our opinion, the cooling/heating performances of real-life water-sorption-based chiller highly depends on the mass/heat transfer performance of the practical systems, and the working performance of EMM-8 will be predominant if the mass and heat transfer properties are well considered. We have added some discussions after the SCP evaluation of the materials.

Changes add to manuscript:

P18: The SCP_{80%} value of EMM-8 surpasses those of both commercial water sorbents and promising MOFs (**Supplementary Table 11**), suggesting a unique opportunity to construct a compact and lightweight sorption system using this adsorbent under the premise of the well arrangement of mass and heat transfer in the packed bed.

12. Describe the key limitations of this study and how they are expected to be addressed.

Answer: Thanks for your constructive comments. In this work, we proposed an SFO-type aluminophosphate as efficient water adsorbent that shows a record COP for ultra-low-temperature-driven sorption cooling. The results indicate a promising application potential of this material for the exploitation of low-grade heat with the superiorities on scalable synthesis, hydrothermal stability, cost, and water sorption performance. Some present limitations

of this work including: i) the hydrothermal synthesis in fluoride medium used to prepare this proposed material is not green enough; and ii) the superior performances and the huge potential of this proposed sorbent stem from the material-scale evaluations, which needs validation by a real-life prototype test. To address the above-mentioned issues, further attempts to develop a green, simple, fast, and safe route for synthesizing this AIPO can be developed to replace the current methods in the future work. Actually, our group is trying to fabricate this SFO-type AIPO via the greener methods, including ionothermal synthesis and the reuse of the organic templates. We are also trying to carry out the 100 g-scale synthesis of this material and its performance evaluation in a large-scale prototype. We will report the results timely in the further works. We have added some discussions after the cost evaluation of the materials.

Changes add to manuscript:

P18: The $SCP_{80\%}$ value of EMM-8 surpasses those of both commercial water sorbents and promising MOFs (**Supplementary Table 11**), suggesting a unique opportunity to construct a compact and lightweight sorption system using this adsorbent under the premise of the well design of mass and heat transfer in the packed bed.

Although the autoclave synthesis in fluoride-containing medium in this work may be a limitation of this material, methods of ionothermal synthesis, reuse of the organic templates can be employed to synthesize this sorbent in more environment-friendly way.

Reviewer 2: The manuscript reports a refrigeration system based on water adsorption, which is built using novel porous materials fabricated by the authors. These materials, zeolite-like porous aluminophosphates, provide high water uptake and help reaching out COP of 0.85, which is quite high for adsorption refrigerators (not sure if it a record though). This result looks very promising for potential application and is worth reporting in a high-profile journal. The paper is nicely written, and I read it with a great interest. Much of the details of the experimental procedures are provided, which would be probably sufficient to reproduce the results (not so much for the computational part). The conclusions are mostly supported by the data, etc. Overall, I would say the paper looks like a good fit to Nature Communications. However, I believe that several things should be taken care of before recommending the manuscript for publication. I am especially concerned with the comment #1 below. The rest, while being important, are just technical.

Reply: The authors would like to thank the reviewer for carefully evaluating the manuscript. More details of the experimental and computational procedures have been added in the revised manuscript, and the technical revisions have also been performed to solve the concerns raised by the reviewer. The provided very detailed review comments allowed us to improve the manuscript. Please find point-to-point answers to comments in the following.

In this work, we proposed and compared the cooling and heating performance of this proposed material with the best current state-of-the-art materials, as shown in the **Supplementary Table4**. As can be seen, the material proposed in this work exhibits the unprecedentedly high COP of 0.85 at driven temperature of 65 °C and evaporation temperature of 5°C, outperforming the other sorbents. Considering the excellent COP, fast water sorption kinetics, and specific energy capacity for cooling at low driven temperature, in our opinion, the proposed material in this work shows the record performance to date.

Major:

1. Please include the details on the regeneration procedure. In fact, use of HF for the regeneration is a concern, which might limit the practical use of the proposed materials. This issue should be discussed in detail.

Answer: Thanks for the meaningful comment from reviewer. We have added some more details on synthesis and activation of the proposed material in the revised manuscript. Typically, 0.662 g pseudo-boehmite as aluminum precursor, and 0.682 ml 85% phosphoric acid as phosphorus source, was slowly added to a solution of 3.6 ml deionized water. After strong stirring at room temperature for 10 min, 0.112 ml of mineralizer 40% HF and 1.234 g 4-dimethylaminopyridine as structure directing agent were added to the mixture. After another 10 min of stirring and 10 min of ultrasonication, the gel was transferred to a 50 ml Teflon-lined stainless-steel autoclave. The reactor was heated and kept at 175 °C for 72~84 h. The white powder was collected by centrifugation, washed with deionized water, and dried in a vacuum oven. Before adsorption characterization, calcination was carried out at 600 °C for 6 h to remove the structure directing agent in the pores of this material. After water adsorption, this AIPs is regenerated at temperature above 65 °C.

In this work, EMM-8 with a record high COP for adsorption refrigerators is prepared in

HF-medium. In fact, HF (or NH₄F) is usually used as a mineralizer to prepare microporous materials including AlPOs with highly crystallinity, which affects the crystallization rate. Undeniably, the use of HF for this synthesis increased the cost of this material due to the corrosiveness and toxicity. For example, the corrosion-resistant reactor, like Teflon-lined stainless-steel autoclave, should be used. The further attempts will be performed, i.e., fluorine-free synthesis and other green routes to prepare this proposed material will be developed in our further works. Additionally, several green methods, such as solvent-free synthesis, dry gel conversion, microwave-assisted synthesis, ionothermal synthesis, and recycling the organic templates, have been used to successfully obtain the AlPOs, as reviewed in the literature²⁰ (*Chem. Rev.* 114, 1521–1543 (2014)). These attempts provide the environment-friendly and cost-effective way to obtain the porous materials, which may further enhance the practical use of the proposed materials. Our group is trying to fabricate SFO-type AlPO via one or some of the above-mentioned methods. We would like to report the results timely in the further works. We have added some discussions about the green synthesis of the materials.

Changes add to manuscript:

P18: Although the autoclave synthesis in fluoride-containing medium in this work may be a limitation of this material, methods of ionothermal synthesis, reuse of the organic templates can be employed to synthesize this sorbent in more environment-friendly way.

P20: The aluminum precursor, 0.662 g pseudo-boehmite, was slowly added to a mixture solution of 3.6 ml deionized water and 0.682 ml 85% phosphoric acid. The solution was strongly stirred at room temperature for 10 min to form a homogeneous suspension. Then, 0.112 ml of mineralizer 40% HF and 1.234 g 4-dimethylaminopyridine were added to the mixture. After another 10 min of stirring and 10 min of ultrasonication, the gel was transferred to a 50 ml Teflon-lined stainless-steel autoclave. The reactor was heated and kept at 175 °C for 72~84 h under static conditions. After cooling down to room temperature, the white powder of EMM-8 was collected by centrifugation, washed with deionized water, and dried in a vacuum oven. The synthesized samples were activated by calcination at 600 °C for 6 h with heating rates of 5 °C·min⁻¹ to remove the SDA. Three samples obtained by repetitive fabrication progresses were used to demonstrate the consistency of the synthesis, as shown in **Supplementary Fig. 20**.

2. Please include a table with all the values needed to calculate the COP and all the equations which are needed for it, so that one can easily follow.

Answer: Thanks very much for the useful suggestion from the reviewer. We have added more information about the methods to calculate COPs in the **Supplementary Note 1-3**, and we provide a table (**Table. R4**) to clearly show all the equations and values needed to calculate the COP. We confirm that these calculation results can be easily reproduced based on the information.

Changes add to Supplementary Information:

P2-8: **Supplementary Note 1-2:** We have added more information about the methods to calculate COPs.

P33-35: **Table. R4** was added to **Supplementary Information** as **Supplementary Table1.**

Table R4. All the values and equations needed for calculating COP

Steps	Purpose	Values needed for calculate	Equations needed for calculating	Remarks
1	Characteristic curve (A-W)	$q(P, T), P, T$	$A(p, T) = -RT \ln \frac{p}{p_0(T)}$ $W = \frac{q(P, T)}{\rho(T)}$	(A-W) is assumed to be temperature-independent
2	Adsorption enthalpy curve ($W - \Delta_{ads} H_w$)	T_1, T_2, T_3 & P_1, P_2, P_3 (W=const)	$\Delta_{ads} H_w = R \left(\frac{\partial \ln P}{\partial (1/T)} \right)_w$	Depend on sorption isotherms at different T
3	Maximum and minimum load (W_{max} & W_{min})	$T_{ev}, T_{con}, T_{des},$ P_{ev}, P_{con}, P_{des}	$A(W_{max}) = -RT_{con} \ln \frac{P_{ev}}{P_0(T_{con})}$ $A(W_{min}) = -RT_{des} \ln \frac{P_{con}}{P_0(T_{des})}$ Characteristic curve (A-W)	Depend on characteristic curve and adsorption potential at W_{max}/W_{min}

Continued Table R4.

Steps	Purpose	Values needed for calculate	Equations needed for calculating	Remarks
4	T_2, T_3	$T_{ev}, T_{con}, T_{des},$ P_{ev}, P_{con}, P_{des}	$A(W_{max}) = -RT_{con} \ln \frac{P_{ev}}{P_0(T_{con})} = -RT_2 \ln \frac{P_{con}}{P_0(T_2)}$ $A(W_{min}) = -RT_{des} \ln \frac{P_{con}}{P_0(T_{des})} = -RT_3 \ln \frac{P_{ev}}{P_0(T_3)}$	Point I and II are linked by W_{max} , point III and IV are linked by W_{min}
5	$Q_{sorption}$	$W_{max}, W_{min}, \rho_{liq}^{wf}, M_w$ Adsorption enthalpy curve $(W - \Delta_{ads} H_w)$	$Q_{sorption} = \frac{1}{M_w} \int_{W_{min}}^{W_{max}} \rho_{liq}^{wf} \Delta_{ads} H(W) dW$	Adsorption enthalpy integral between W_{max} and W_{min}
6	Q_{ev}, Q_{con}	$\Delta_{vap} H(T_{ev}), \Delta_{vap} H(T_{con}),$ $m_{sorption}, \Delta W$	$Q_{ev} = - \frac{\Delta_{vap} H(T_{ev}) \rho_{liq}^{wf} m_{sorbent} \Delta W}{M_w}$ $Q_{con} = \frac{\Delta_{vap} H(T_{con}) \rho_{liq}^{wf} m_{sorbent} \Delta W}{M_w}$ $\Delta W = W_{max} - W_{min}$	Q_{ev} is positive, Q_{con} is negative

Continued Table R4.

Steps	Purpose	Values needed for calculate	Equations needed for calculating	Remarks
7	The energy for each stage Q_{I-II} , Q_{II-III} , Q_{III-IV} , Q_{IV-I}	T_{con} , T_2 , T_3 , T_{des} , W_{max} , W_{min} , $\rho_{liq}^{wf} c_p^{wf}(T)$, $Q_{sorption}$	$Q_{I-II} = \int_{T_{con}}^{T_2} c_p^{eff}(T) dT + \int_{T_{con}}^{T_2} \rho_{liq}^{wf} W_{max} c_p^{wf}(T) dT$ $Q_{II-III} = \int_{T_2}^{T_{des}} c_p^{eff}(T) dT + \int_{T_2}^{T_{des}} \rho_{liq}^{wf} \frac{W_{max} + W_{min}}{2} c_p^{wf}(T) dT - Q_{sorption}$ $Q_{III-IV} = \int_{T_{des}}^{T_3} c_p^{eff}(T) dT + \int_{T_{des}}^{T_3} \rho_{liq}^{wf} W_{min} c_p^{wf}(T) dT$ $Q_{IV-I} = \int_{T_3}^{T_{con}} c_p^{eff}(T) dT + \int_{T_3}^{T_{con}} \rho_{liq}^{wf} \frac{W_{max} + W_{min}}{2} c_p^{wf}(T) dT + Q_{sorption}$ $c_p^{eff} \approx c_p^{sorption} \approx 1.0 J \cdot g^{-1} \cdot K^{-1}$	Q_{I-II} , Q_{II-III} is positive, Q_{III-IV} , Q_{IV-I} is negative
8	Q_{regen} , Q_{ads}	Q_{I-II} , Q_{II-III} , Q_{III-IV} , Q_{IV-I}	$Q_{regen} = Q_{I-II} + Q_{II-III}$ $Q_{ads} = Q_{III-IV} + Q_{IV-I}$	Q_{regen} is positive, Q_{ads} is negative
9	COP	Q_{ev} , Q_{con} , Q_{regen} , Q_{ads}	$COP_H = -\frac{Q_{con} + Q_{ads}}{Q_{regen}}$ $COP_C = \frac{Q_{ev}}{Q_{regen}}$	$1 \leq COP_H \leq 2$ $COP_C \leq 1$

2. Details of GCMC simulations: what code was used? what water model was used? what EOS was used to relate chemical potential to pressures?

Answer: Thanks for the useful comment from the reviewer. We performed GCMC simulations by using Sorption code in commercial software of Material Studio 19.1. The simple SPC water model was used to describe the water–water interactions for EMM-8. The interactions between the guest water molecules and the MOF structure were described by a combination of site-to-site LJ contributions and Coulombic terms. The augmented CVFF force field parameter was adopted to describe the LJ parameters for the atoms in the AIPOs framework. A cutoff radius was set to 1.2 nm for the LJ interactions, while the long-range electrostatic interactions were handled by the Ewald summation technique. We have added more details of GCMC simulations, including the code we used, the LJ-parameters used for simulation of water adsorption, and the progress we obtain the atomic charges of the AIPOs in the revised manuscript and the Supplementary Information.

Changes add to manuscript:

P23-24: The framework structure was constructed from their corresponding experimental single-crystal diffraction data. First-principles calculations were first performed within the framework of density functional theory (DFT) and the plane-wave pseudopotential functional approach, as implemented in the Vienna ab initio Simulation Package (VASP). Ion-electron interactions were implemented with the projector-augmented wave (PAW) method. The generalized gradient approximation (GGA) and the PBE functional⁵³ were used with a 450 eV limit for the plane-waves. All the atom positions were relaxed until the energy and force changes on each atom were less than 0.001 meV and $0.01 \text{ eV}\cdot\text{\AA}^{-1}$, respectively. For geometry optimization and electronic structure calculations, the Brillouin zone was sampled with $2 \times 4 \times 4$ Γ -centered Monkhorst-Pack K-point grid. Based on the self-consistent charge and potential, the Density Derived Electrostatic and Chemical (DDEC) net atomic charges are calculated (**Supplementary Table 13**) using chgemo code. Grand canonical Monte Carlo (GCMC) simulations were employed by Sorption code in commercial software of Material Studio to calculate the water adsorption in the AIPO at 303 K. The material was treated as rigid frameworks with atoms frozen at their crystallographic positions, and the simulation box was made of 12 conventional unit cells ($2 \times 2 \times 3$). A cutoff radius was set to 1.2 nm for the LJ interactions, while the long-range electrostatic interactions were handled by the Ewald summation technique. The augmented CVFF force field parameter was adopted to describe the LJ parameters for the atoms in the AIPOs framework, whereas water molecules were described by the simple SPC potential model (**Supplementary Table 14**). This combination has been proven to be effective for the calculation of water sorption isotherm in zeolite. Periodic boundary conditions were considered in all three dimensions. For each state point, GCMC simulations consisted of 2×10^8 steps to ensure the equilibration, followed by 3×10^8 steps to sample the desired thermodynamic properties. In addition, to obtain accurate ensemble averages in GCMC simulations, at least millions of configurations generated by random translation, rotation, regrowth, and swap moves are sampled in each simulation.

Changes add to Supplementary Information:

P50: **Supplementary Table 14** was added to **Supplementary Information**.

Supplementary Table 14. LJ potential parameters for the atoms of the H₂O, and EMM-8.

Elements	σ (Å)	ϵ (Kcal·mol ⁻¹)	q(e)
Al	2.941	9.043	List in
P	4.20	0.200	Supplementary
O	3.210	0.228	Table 13
OW	3.166	0.155	-0.820
HW	0	0	+0.410

4. Computational data: all the DFT and GCMC data (input files and equilibrated configurations) should be documented as supporting data to secure the reproducibility

Answer: Thanks for the useful comment from the reviewer. The DFT and GCMC simulations were performed using two commercial softwares, VASP and Material Studio. The input files mainly include the framework structure file of the AIPO, the Force-Field Parameters, and the DDEC net atomic charges used in this work. The framework structure file of the AIPO was obtained from the literature **ref. 21** (*Acta Cryst. B*63, 56-62 (2007)), which was cited in the manuscript. The Force-Field Parameters and the DDEC net atomic charges used in this work have been list in **Supplementary Table 13** and **14**. Moreover, all configurations needed to reproduce these calculations were included in the Method. Please see **Answer 3**. We believe that one can easily repeat these calculations based on the supporting information.

Minor:

SI P4: indexes are missing for

Answer: Thanks for the useful comment from the reviewer. As your suggestion, we checked the indexes on page 4 in **Supplementary Information**. The format of indexes on page 4 was wrong, and it actually referred to **Supplementary Reference 1**. We have improved these two parts, **Supplementary Note 1** and **Supplementary Note 2**, and corrected the reference format mentioned in **Supplementary Note 2** on page 5.

SI Fig 1(a,b): hard to read because of low quality/resolution

Answer: Thanks for the useful comment from the reviewer. We have checked and improved the resolution of **Supplementary Figure 1**, as shown in the following figure.

Supplementary Figure 1. Schematic diagram of typical adsorption heat transfer device.

SI Fig 7(d): scale is not given

Answer: Thanks for the useful comment from the reviewer. We have given the diameter of tablet shaped by EMM-8 in **Supplementary Figure 7d**, as shown in the following figure.

Supplementary Figure 8. Typical characterization of scaled-up samples. a. SEM image of the EMM-8 prepared by 2L-scale Teflon-lined stainless-steel autoclave. **b.** Comparison of XRD patterns between the EMM-8 prepared by 2L-scale and 50ml-scale. **c.** The nitrogen adsorption-desorption isotherms of EMM-8 prepared by 2L-scale and 50ml-scale. The micropore size distribution was shown in the insert map. **d.** Photo of shaped EMM-8.

SI Fig 4, 13: T units are missing [same in other figures]

Answer: Thanks for your comment. As your suggestion, we have checked the T units (°C) in all

pictures in **Supplementary Information** file, as **Supplementary Figure 4, 13, 14, 15** etc.

SI Fig 16: hard to read because of low quality/resolution

Answer: Thanks for the useful comment from the reviewer. We have checked and improved the resolution of **Supplementary Figure 16**, as shown in the following figure.

Supplementary Figure 17. View and scheme of self-constructed apparatus for the test of water adsorption kinetics installed at Research Center of Solar Power & Refrigeration labs in Shanghai Jiao Tong University.

Reviewer 3:

The authors report on a performance of porous aluminophosphate with SFO topology in water-sorption driven heat transformation (refrigeration). The manuscript is very well written and the study of the material well performed. I am very glad that the authors recognized the potential of aluminophosphates in heat transformation applications and also tried to address one of their crucial drawbacks, which is the cost of materials (due to the use of mostly costly structure directing agents). At the same time they emphasized the structure stability of aluminophosphates in humid conditions and a nice attempt to scale up the synthesis. However, there are some points which need to be addressed, before the manuscript is considered for publication in Nature Communications.

Reply: The authors would like to thank the reviewer for carefully evaluating the manuscript and for giving positive comments on our work. The provided very detailed review comments allowed us to improve the manuscript. Please find point-to-point answers to the reviewer's comments in the following.

1. The authors compare ALPO-SFO with MOF materials and at some points also with SAPO-34 (aluminophosphate doped with Si). I think that more detailed comment on/comparison with pure aluminophosphates, like ALPO-TRIC (Ristic et al, *Advanced Functional Materias*, 2012, not cited in the manuscript) and ALPO-LTA (Krajnc et at., 2017, cited in the Introduction and mentioned in Figure 5, Supp Fig. 15), would be appropriate. It is true that both listed materials were mostly evaluated for water-sorption-based heating applications (ALPO-LTA was evaluated for cooling in conference contribution Eurosun, 2018, Ristic et al.), but the basic principles are the same. Furthermore, they are all pure aluminophosphates and not Si-doped version (doping can crucially change sorption characteristics in low pressure range). Additionally, the ALPO-SFO, ALPO-TRIC and ALPO-LTA were all prepared in F-media (which adds to relaxation of the SBU) and it is anticipated that this also contributes to final water sorption mechanism of activated material. How do the authors explain the selection of materials to compare with and how they explanation the obtained sorption mechanism (S-shape isotherm) of their material? I am quite sure, it is not only pore opening size.

Answer: Thanks for your constructive and meaningful comments. Comprehensive investigations of water sorption on AIPOs (especially pure aluminophosphates without Si or metal doping) are lack in comparison to those of MOFs, which limit the design of aluminophosphates for application where water is involved. We greatly appreciate the recommendation of APO-TRIC²² and ALPO-LTA from referee that help us to deeply evaluate ALPO-SFO and understand their structure–property relationship. We have added the detailed comparison between ALPO-SFO and other AIPOs (ALPO-TRIC and ALPO-LTA), as shown in **Figure R9** and **Table R5**. It can be seen that these pure aluminophosphates show perfect S-type water adsorption isotherms (**Figure R9**), indicating the similar water sorption mechanism, forming of ordered hydrogen-bonded water clusters followed by the initial bonding of water molecules to Al sites. In comparison of these results with water sorption measurement results of SAPO-34 and several MOFs reveals that excessive hydrophilic adsorption sites shift the water uptake to even lower relative pressure. However, the heats of adsorption for those three AIPOs are different. ALPO-SFO exhibits the lower average adsorption enthalpy of 46.8 kJ·mol⁻¹ than that of ALPO-TRIC (53.6 kJ·mol⁻¹) and ALPO-LTA (55.6 kJ·mol⁻¹),

which contributes to higher COP for ALPO-SFO (Table R5). In our opinion, it may be attributed to the various pore sizes. The interaction between water molecules and the framework in 12-ring windows of ALPO-SFO is weaker than that of ALPO-TRIC and ALPO-LTA with 8-ring openings. This comparison provides guidance to understand the structure–property relationship and thus to design the aluminophosphates for heat transformation application. We have added some discussions in the last paragraph of Section Water adsorption evaluation to state these comparisons.

Figure R9. Water-adsorption isotherms for EMM-8 (○) and other AIPOs & SAPOs adsorbents: AIPO-LTA (○), AIPO-5 (○), AIPO-Tric (○), and SAPO-34 (○)

Table R5. Water sorption properties and heat storage capacity of EMM-8 and other AIPOs & SAPOs adsorbents.

Material	Crystal density (g cm ⁻³)	V _{mic} ^a (cm ³ g ⁻¹)	<-Δ _{ads} H> (kJ mol ⁻¹)	Working capacity ^b		Heat storage capacity ^c	
				(g g ⁻¹)	(ml ml ⁻¹)	(Wh kg ⁻¹)	(kWh m ⁻³)
EMM-8	1.50	0.30	46.76	0.274	0.411	212	318
AIPO-LTA	1.41	0.32	55.58	0.360	0.508	319	450
AIPO-5	1.75	0.15	56.10	0.176	0.307	163	286
SAPO-34	1.43	0.212	57.00	0.178	0.255	168	240
AIPO-Tric	/	/	53.60	0.275	/	/	/

^a Based on N₂ sorption isotherms at 77 K.

^b Working capacity deduced from one refrigeration cycle at T_{ev} = 10 °C, T_{con} = 30 °C, and T_{des} = 80 °C;

^c Energy storage capacity per unit weight or volume of adsorbent at T_{ev} = 10 °C, T_{con} = 30 °C, and T_{des} = 80 °C;

HF (or NH₄F) is usually used as a mineralizer to prepare AIPOs with highly crystallinity, which affect the crystallization rate. Fluorine is coordinated to the penta-coordinated Al sites in the as-prepared ALPO-SFO and ALPO-TRIC, but after calcination fluorine atoms are removed from the structure and all Al atoms become tetrahedrally coordinated, which is demonstrated by ¹⁹F and

²⁷Al MAS NMR studies in the previous work of Ristic et al²² (*Adv. Funct. Mater.* 22, 1952–1957 (2012)) and Afeworki et al²³ (*Microporous Mesoporous Mater.* 103, 216-224 (2007)). The calcined ALPO-SFO samples only exhibit tetrahedrally coordinated Al atoms, which is also confirmed by ²⁷Al MAS NMR and XPS in our work. Ristic et al²² found that slightly disordered structure formed after removal of fluorine, which is a driving force for water sorption in the APO-Tric structure and also determines the perfect ordering of the second layer of water molecules. It is worthy noting that the ALPO-SFO can also be prepared in a fluoride-free medium as reported by Cao et al²¹ (*Acta Cryst.* B63, 56-62 (2007)). We would like to evaluate the difference of water adsorption between the obtained ALPO-SFO with and without HF in the further work. We have added the citation of the results of Ristic et al²² in the Introduction to state their finding.

As discussed in this work and the previous studies, the pore filling mechanism, i.e., the formation of ordered hydrogen-bonded water clusters followed by the initial bonding of water molecules to hydrophilic adsorption sites, well explaining the S-shaped isotherm. However, we think that how the structural parameters influencing the obtained sorption mechanism is not very clear, and comprehensive and predictive model of water adsorption is lack to date. Based on the existing investigations, we try our best to explain the structure–property relationship. First, to obtain the S-type isotherm, excess hydrophilic adsorption centers should be avoided that can adsorb more water molecular at the lower relative pressure. A typical example is SAPO-34, which exhibit the gradual water uptake due to the presence of the formation of highly acidic bridging OH groups (Si–OH–Al). Also, the pore opening sizes play the important roles in water adsorption. The step-wise location of isotherm and the heat of adsorption are influenced by the pore size as discussed in many previous works^{24,25} (*Chem. Soc. Rev.* 43, 5594-5617 (2014); *J. Phys.: Condens. Matter.* 22, 284106 (2010)). Moreover, electrostatic interactions are an important factor due to the fact that water is the highly polar molecular, as shown in Figure 3b. Therefore, zeolites X, Y and Na-rho-ZMOF with an anionic framework and charge-balancing nonframework ions show the high water uptake at very low relative pressure (<0.01), which causes a high desorption temperature and heat of adsorption. In summary, the S-type isotherm of AIPO-SFO and its corresponding high energy performance in adsorption cooling can be ascribed to the appropriate pore size, suitable adsorption sites, and electrically neutral frameworks. We have added some discussions after the GCMC analysis to address these statements.

Changes add to manuscript:

P4-5: In 2012, Ristić et al reported the first test of APO-Tric for water sorption applications. This microporous aluminophosphate shows S-shaped water adsorption isotherm and excellent water-sorption-based heat storage performance. The water adsorption mechanism of this AIPO was revealed that the formation of highly ordered H-bond water cluster is followed by the initial water sorption on Al, previously coordinated to two F in the APO-Tric.

P13-14: The pore-filling mechanism of water uptake mentioned above is also observed in other AIPOs, such as AIPO-LTA, AIPO-Tric, and AIPO-18, suggesting the similar stepwise isotherms of these aluminophosphates (Supplementary Fig. 9b). The exception is SAPO-34. The incorporation of Si, leading to the formation of highly acidic bridging OH groups as rather strong adsorption centers for water molecules, are associated with high water uptake at low relative pressure. However, EMM-8 shows lower heat of adsorption than other reported AIPOs, as shown Supplementary Table 5. This is mainly due to the weaker water–framework interactions and the

weaker interaction between the water molecules contributed by the larger cavity of EMM-8 with 12-ring openings than that of AIPO-LTA and AIPO-Tric with 8-membered rings, which is illustrated by Demontis et al. We could reveal the structure-property relationship, where appropriately weaker water adsorption sites and properly larger pore sizes of EMM-8 result in the low adsorption enthalpy, providing the guidance to design the porous materials for water sorption applications.

Changes add to Supplementary Information:

P19: **Fig. R9** was added to **Supplementary Information** as **Supplementary Figure 9b**.

2. S-shape isotherm is mentioned as optimal type for selected applications. This “requirement” is many times misinterpreted in the literature. It is considered beneficial, because it indicates that material would be dehydrated at lower T. However, we could conclude it from de-sorption part. Sometimes, there is a strong hysteresis between adsorption and desorption curves. Could authors provide de-sorption isotherm?

Answer: Thanks for the useful suggestion from the reviewer. Strong hysteresis during desorption is undesired, as it will increase the required desorption temperature. We have added the complete water adsorption-desorption isotherms of AIPO-SFO at 30 °C, as shown in **Figure R10**. A slight hysteresis is observed in the desorption branch in the narrow range of $P/P_0 = 0.1-0.15$, which may be attributed to the presence of the mesoporous. Hence, we think the water desorption for AIPO-SFO could be driven at low temperature. This conclusion is confirmed by the experimental measurements of water adsorption/desorption kinetics, as exhibited in **Figure 5f**. The water desorption profiles show the fast desorption rates under the operating conditions of $P_{H_2O}=4.24$ kPa, and $T_{des}=65$ °C, demonstrating its superiority in the deep utilization of ultra-low-temperature heat. We have added some statements in the first paragraph of Section water sorption evaluation to describe and analyze the small hysteresis along with the addition of the water adsorption-desorption isotherms at 30 °C in **Supplementary Figure 9 c**.

Figure R10. Adsorption/desorption isotherms for EMM-8 at 30 °C

Changes add to manuscript:

P11: A small hysteresis is observed in the desorption branch in the narrow range of $P/P_0 = 0.1-0.15$ (Supplementary Fig. 9c), which may be attributed to the presence of the mesoporous.

Changes add to Supplementary Information:

P19: Fig. R10 was added to Supplementary Information as Supplementary Figure 9c

3. Structure stability in humid conditions; do authors consider 24 h test at 100 deg.C appropriate/long enough? Cycling stability test, which is relevant for application—it is mentioned in the manuscript, but more information on the procedure are needed? Comment—thermal stability up to 700 deg. C is compared with MOFs, but ALPO-TRIC is thermally stable up to 900 deg. C.

Answer: Thanks very much for the useful comments from the reviewer. Cycling stability of AIPOs in humid conditions is a key point for their applications where water is involved. Several previous investigations on hydrothermal stability of AIPOs have been reported. According to the work by Bauer et al.²⁶ (*Stud Surf Sci Catal.* 837-844 (2007)), the silicon content is a crucial factor for the stability of SAPO-34 in water vapor atmosphere. The high silicon containing SAPO-34 sample undergo a fast irreversible structural degradation, while the low silicon sample was proved to undergo complete reversible structural changes. Fischer²⁷ (*Phys. Chem. Chem. Phys.* 18, 15738–15750 (2016)) found that the difference of topologies is relate to the stability of AIPOs in humid conditions. The materials with GIS and RHO topologies exhibit drastically distorted structure when water molecules are desorbed. In our work, a harsh condition, i.e., 24 h at 100 °C, is used to test the hydrothermal stability of AIPO-SFO. The results point out that the sample can maintain its crystallinity, framework structure, and microporosity during hydrothermal treatments, demonstrating the huge application potential of AIPO-SFO for water adsorption cooling. Although more times cycling stability tests are always more reliable to verifying the stability, we think that the test conditions and times used in this work were proper to evaluate hydrothermal stability of water sorbents in this stage due to negligible performance degradation. Our group is trying to carry out the 100 kg-scale synthesis of this material and its performance evaluation in a large-scale prototype. Cycling tests for a long time are considered to evaluate the stability of AIPO-SFO in depth. We would like to report the results timely in the hope of promoting the application of AIPOs for water-sorption cooling.

We have added more information about cycling stability test in the section Method of the revised manuscript. Multiple cycles of water adsorption-desorption were also performed using a 3H-2000 PW intelligent gravimetric analyzer. Hydration/dehydration cycling performance was carried out with the sequential procedure of isothermal measurements at 30 °C and relative humidity of 0.3 until constant weight, followed by drying at 110 °C to constant weight under high vacuum. The sequence was repeated 30 times.

In general, thermal stability of AIPOs is better than that of MOFs, which is an obvious advantage of AIPOs. We are grateful to the reviewer for the information about ALPO-TRIC with higher thermal stability. We have added the citation of this work to further state the superiority of AIPOs in thermal stability.

Changes add to manuscript:

P9: Temperature-dependent powder XRD (**Fig. 1a**) and thermogravimetric analysis (TGA) (**Supplementary Fig. 5**) reveal the total removal of the SDA at ~400 °C and demonstrate the desirable thermal stability of the synthesized EMM-8 up to 700 °C, which is slightly lower than that of AlPO-LTA and AlPO-Tric, but much better than that of MOFs.

P22: Multiple cycles of water adsorption-desorption were also performed using a 3H-2000 PW intelligent gravimetric analyzer. Hydration/dehydration cycling tests were carried out with the sequential procedure of isothermal measurements at 30 °C and relative humidity of 0.3 until reaching constant weight, followed by drying at 110 °C to a constant weight under high vacuum, which was repeated 30 times. In prior to the multiple cycle experiment, the first cycle was carried out by a different condition: EMM-8 is dehydrated at 150 °C overnight under vacuum, hydrated at 30 °C and 30% RH, and then dehydrated again at 110 °C under high vacuum.

4. Price of material (Supp. Table 9); Did authors consider the cost of autoclave synthesis vs. autoclave-free synthesis (MOFs mostly). Can the authors comment the IL synthesis of ALPOs, which is also considered greener?

Answer: Thanks for the meaningful comments from the reviewer. In this work, only the cost of raw materials used to prepare the sorbents is considered as seen in **Supplementary Table 12**. It can be seen that the AlPO-SFO shows much cheaper price than MOFs and AlPO-LTA, indicating its extremely attractive advantage in commercialization potentials. We did not consider the effect of synthesis methods on the cost in this present work because of the huge room of optimization. Although the autoclave synthesis is used to prepare the AlPO in this work, we think that there are a lot of methods to make the synthesis cheaper and greener, as reviewed in literature **ref. 20** (*Chem. Rev.* 114, 1521–1543 (2014)). For example, solvent-free synthesis, dry gel conversion, microwave-assisted synthesis, ionothermal synthesis, and recycling the organic templates have been used to successfully obtain the AlPOs. Our group is trying to fabricate AlPO-SFO via the above-mentioned methods. We would like to report the results timely in the further works.

For your mentioned ionothermal synthesis, it has obvious advantages, such as the following: (i) ILs can act as both solvent and template as well as charge-compensation groups; (ii) ionothermal synthesis can be performed at ambient pressure; and (iii) ILs offer a novel chemical environment for organic and inorganic reactants. Several successful examples of ionothermal synthesis of AlPOs, such as LTA, PHI, GIS, and MER structure types, have been reported. In our previous work¹⁴ (*Microporous Mesoporous Mater.* 305, 110315 (2020)), AlPO-LTA with uniform size and cubic morphology were obtained by ionothermal synthesis method with optimized parameters and two-step calcination template removal, which shows significant stability in extreme conditions, high micropore volume of 0.32 ml·g⁻¹, and excellent water adsorption capacity of 0.38 g·g⁻¹ at $P/P_0=0.20$. Moreover, the combination of ionothermal synthesis and microwave heating is considered a simple, fast, environmentally benign, and safe route for synthesizing AlPOs. However, the expensive cost of ILs is a big obstacle that should be considered.

We have added some discussions after the cost analysis of the materials, which we hope meet with approval.

Changes add to manuscript:

P18: Although the autoclave synthesis in fluoride-containing medium in this work may be a limitation of this material, methods of ionothermal synthesis, reuse of the organic templates can be employed to synthesize this sorbent in more environment-friendly way.

5. I guess the $0.36 \text{ g}\cdot\text{g}^{-1}$ water uptake is maximal water uptake at $p/p_0=0.8$. What is the uptake at the S-step (applicable for application)?

Answer: Thanks for the meaningful comments. As the reviewer's mentioned, the maximal water uptake of $0.36 \text{ g}\cdot\text{g}^{-1}$ is observed at high relative pressure. The proposed AIPO exhibits a relatively high water adsorption capacity of $0.25\text{-}0.28 \text{ g}_{\text{H}_2\text{O}}\cdot\text{g}_{\text{sorbent}}^{-1}$ at $25 \text{ }^\circ\text{C}$ and $P/P_0 = 0.16\text{-}0.2$, which is $0.21\sim 0.24 \text{ g}_{\text{H}_2\text{O}}\cdot\text{g}_{\text{sorbent}}^{-1}$ different from $0.039 \text{ g}_{\text{H}_2\text{O}}\cdot\text{g}_{\text{sorbent}}^{-1}$ at $P/P_0 = 0.15$, as shown in the **Figure R11**. We have corrected the description in Abstract about water uptake.

Figure R11. Water adsorption isotherm of EMM-8 at $25 \text{ }^\circ\text{C}$

Changes add to manuscript:

Abstract: The EMM-8 is characterized by 12-membered ring channels with large accessible pore volume and exhibits high water uptake of $0.28 \text{ g}\cdot\text{g}^{-1}$ at $P/P_0=0.2$, low-temperature regeneration of $65 \text{ }^\circ\text{C}$, fast adsorption kinetics, remarkable hydrothermal stability, and scalable fabrication.

6. Page 4: there is at least one more aluminophosphate developed, i.e. AQSOA-Z05 at the beginning (in 2005).

Answer: Thanks very much for the useful suggestion from the reviewer. AQSOA-Z05, which is also developed by Mitsubishi Plastics from Japan, is usually used as desiccant for humidity conditioning owing to its easy regeneration at low temperature. We have added it in the revised manuscript.

Changes add to manuscript:

P4: Several years ago, the company of Mitsubishi Plastics from Japan developed three types of aluminophosphates (namely AQSOA-Z01 AQSOA-Z02 and AQSOA-Z05), representing a

considerable advancement for ADC applications owing to their outstanding performances of high water uptake and low regeneration temperature (< 90 °C).

7. Page 5, end of first paragraph: Could authors just shortly explain which types and characteristics of MOFs, that are better than some already mentioned aluminophosphates, for example ALPO-LTA.

Answer: Thanks for the useful suggestion from reviewer. The recent reported MOFs, Co-CUK-1 and KMF-1, shows better energy performances than available aluminophosphates including ALPO-LTA. Co-CUK-1 exhibits the high COP of 0.83 (for cooling) and 1.77 (for heating), which is the highest values except for AlPO-SFO of this work. The apparent reason for the notably high energy performance is the lower heat of adsorption of $46.9 \text{ kJ}\cdot\text{mol}^{-1}$ for Co-CUK-1 than ALPO-LTA ($55.6 \text{ kJ}\cdot\text{mol}^{-1}$) due to the fact that the theoretical maximum COP value related with the ratio of water evaporation enthalpy and adsorption enthalpy. This low enthalpy of adsorption may be partly attributed to the larger pore opening size of 1.3 nm for Co-CUK-1 as discussed in **Answer 1**. More investigations on water adsorption mechanism and structure-property relationship are needed to guide the design of AlPOs for water-based sorption chiller.

Changes add to manuscript:

P5: Unfortunately, the AlPOs reported so far cannot hold a candle to the best current MOFs, such as Co-CUK-1 and KMF-1, especially in terms of the performance of ultra-low-temperature-driven chillers. More investigations on screening and design of AlPOs for water-sorption-based heating and cooling applications are urgently needed.

8. Supp. Table 3: Please check the data for ALPO-LTA.

Answer: Thanks for the useful suggestion from reviewer. We checked the thermodynamic calculations methods described in **Supplementary Note 2** and compared the calculated results with the previous literatures, **ref.9** (*Adv. Energy Mater.* 7, 1601815 (2017)) and **ref. 28** (*In Proceedings of the ISES Eurosun 2018 conference-12th International Conference on Solar Energy for Buildings and Industry*, 704-709). We used the same water sorption results of AlPO-LTA with those of Krajnc et al⁹, which is a little different with Ristić et al²⁸, as shown in **Figure R12**. Based on the thermodynamic calculation methods described in **Supplementary Note 2**, we calculated cooling enthalpy Q_{ev} and the working capacities at $T_{des} = 100 \text{ °C}$ based on the sorption data of **ref. 9** and **ref. 28**, and compared them with the curve given in **ref. 9** (**Fig. R13** and **Table R6**). The three curves agree well, suggesting the reliability of our calculation method. Then, we calculated the cooling enthalpy at $T_{des} = 65 \text{ °C}$ (**Fig. R14**) and the working capacity (**Table R7**) determined according to the characteristic curve from **ref. 9** and **ref. 28**. The obvious differences can be observed. Base on this present method, $Q_{ev} = 120 \text{ kWh}\cdot\text{m}^{-3}$ ($306 \text{ kJ}\cdot\text{kg}^{-1}$) at $T_{ev} = 10 \text{ °C}$, $T_{con} = 30 \text{ °C}$, and $T_{des} = 65 \text{ °C}$, as shown in **Supplementary Figure 16** and **Supplementary Table 5**, which is different from the value in **ref. 28**. It may be caused by a large error in determining the minimum load due to the step-wise curve (Pink vertical line in Fig. R1). However, the COP_c at $T_{ev} = 5 \text{ °C}$, $T_{con} = 30 \text{ °C}$, and $T_{des} = 65 \text{ °C}$ calculated by the experimental results of **ref. 9** and **ref. 28** show the similar values, as shown in **Fig. R15**. Therefore, we think the thermodynamic data for the

materials in our work is reliable.

Figure R12. Characteristic curve in ref. 9 and ref. 28.

Figure R13. Cooling enthalpy Q_{ev} in one refrigeration cycle as a function of temperature lift ΔT (T_{con} = 30 °C, T_{des} = 100 °C).

Table R6. The working capacities at different ΔT ($T_{con} = 30\text{ }^{\circ}\text{C}$, $T_{des} = 100\text{ }^{\circ}\text{C}$).

ΔT ($^{\circ}\text{C}$)	T_{ev} ($^{\circ}\text{C}$)	Working capacities ΔW ($\text{ml}\cdot\text{g}^{-1}$)	
		Based on ref. 9	Based on ref. 28
2.5	27.5	0.402	0.406
5	25	0.397	0.400
7.5	22.5	0.391	0.394
10	20	0.385	0.388
12.5	17.5	0.381	0.384
15	15	0.377	0.378
17.5	12.5	0.372	0.373
20	10	0.367	0.369
22.5	7.5	0.363	0.364
25	5	0.359	0.360
27.5	2.5	0.355	0.355
28	2	0.354	0.354

Figure R14. Cooling enthalpy Q_{ev} in one refrigeration cycle as a function of temperature lift ΔT ($T_{con} = 30\text{ }^{\circ}\text{C}$, $T_{des} = 65\text{ }^{\circ}\text{C}$).

Table R7. The working capacities at different ΔT ($T_{con} = 30\text{ }^{\circ}\text{C}$, $T_{des} = 65\text{ }^{\circ}\text{C}$).

ΔT ($^{\circ}\text{C}$)	T_{ev} ($^{\circ}\text{C}$)	Working capacities ΔW (ml \cdot g $^{-1}$)	
		Based on ref. 9	Based on ref. 28
2.5	27.5	0.155	0.204
5	25	0.149	0.199
7.5	22.5	0.143	0.192
10	20	0.138	0.186
12.5	17.5	0.134	0.182
15	15	0.129	0.176
17.5	12.5	0.124	0.171
20	10	0.120	0.167

Figure R15. Coefficient of performance values for cooling calculated basis on ref. 9 and ref. 28 ($T_{ev} = 5\text{ }^{\circ}\text{C}$ and $T_{con} = 30\text{ }^{\circ}\text{C}$)

Finally, we authors thank the editor and reviewers for your good comments and suggestions!

Reference

1. de Lange, M. F., Verouden, K. J. F. M., Vlugt, T. J. H., Gascon, J. & Kapteijn, F. Adsorption-driven heat pumps: the potential of metal–organic frameworks. *Chem. Rev.* **115**, 12205-12250 (2015).
2. Furukawa, H. et al. Water adsorption in porous metal–organic frameworks and related materials. *J. Am. Chem. Soc.* **136**, 4369-4381 (2014).
3. Wang, S. et al. A robust large-pore zirconium carboxylate metal–organic framework for energy-efficient water-sorption-driven refrigeration. *Nat Energy* **3**, 985-993 (2018).
4. Cadiou, A. et al. Design of hydrophilic metal organic framework water adsorbents for heat reallocation. *Adv. Mater.* **27**, 4775-4780 (2015).
5. Reinsch, H. et al. Structures, sorption characteristics and nonlinear optical properties of a new series of highly stable aluminium MOFs. *Chem. Mater.* **25**, 17-26 (2013).
6. Lee, J. S. et al. The porous metal-organic framework CUK-1 for adsorption heat allocation toward green applications of natural refrigerant water. *ACS Appl. Mater. Interfaces.* **11**, 25778-25789 (2019).
7. Cho, K. H. et al. Rational design of a robust aluminum metal-organic framework for multi-purpose water-sorption-driven heat allocations. *Nat Commun* **11**, 5112 (2020).
8. Cho, K. H. et al. Defective Zr-fumarate MOFs enable high-efficiency adsorption heat allocations. *ACS Appl. Mater. Interfaces* **13**, 1723-1734 (2021).
9. Krajnc, A. et al. Superior performance of microporous aluminophosphate with LTA topology in solar-energy storage and heat reallocation. *Adv. Energy Mater.* **7**, 1601815 (2017).
10. Yeh, R. L., Ghosh, T. K. & Hines, A. L. Effects of regeneration conditions on the characteristics of water vapor adsorption on silica gel. *J. Chem. Eng. Data.* **37**, 259-261 (1992).
11. Lenzen D. et al. A metal–organic framework for efficient water-based ultra-low-temperature-driven cooling. *Nat Commun* **10**, 3025 (2019).
12. Rieth, A. J., Yang, S., Wang, E. N. & Dincă, M. Record atmospheric fresh water capture and heat transfer with a material operating at the water uptake reversibility limit. *ACS Cent. Sci.* **3**, 668-672 (2017).
13. Pérez-Carvajal, J., Boix, G., Imaz, I. & Maspoch, D. The imine-based COF TpPa-1 as an efficient cooling adsorbent that can be regenerated by heat or light. *Adv. Energy Mater.* **9**, 1901535 (2019).
14. Liu, Z. L., Xu, M., Huai, X. L., Huang, C. F. & Lou, L. T. Ionothermal synthesis and characterization of AlPO₄ and AlGaPO₄ molecular sieves with LTA topology. *Microporous Mesoporous Mater.* **305**, 110315 (2020).
15. Kim, Y. D., Thu, K. & Ng, K. C. Adsorption characteristics of water vapor on ferroaluminophosphate for desalination cycle. *Desalination.* **344**, 350-356 (2014).
16. Zhou, L. S., Guan, J. K., Yu, C. L. & Huang, B. C. MnOx Supported on Hierarchical SAPO-34 for the Low-Temperature Selective Catalytic Reduction of NO with NH₃: Catalytic Activity and SO₂ Resistance. *Catalysts.* **11**, 314 (2021).
17. Yan, J. et al. Adsorption isotherms and kinetics of water vapor on novel adsorbents MIL-101(Cr)@GO with super-high capacity. *Appl. Therm. Eng.* **84**, 118-125 (2015)
18. Radu, A. I., Defraeye, T., Ruch, P., Carmeliet, J. & Derome, D. Insights from modeling dynamics of water sorption in spherical particles for adsorption heat pumps. *Int. J. Heat Mass Transf.* **105**, 326-337 (2017).

19. Ammann, J., Michel, B., Studart, A. R. & Ruch, P. W. Sorption rate enhancement in SAPO-34 zeolite by directed mass transfer channels. *Int. J. Heat Mass Transf.* **130**, 25-32 (2019).
20. Meng, X. J. & Xiao, F. S. Green Routes for Synthesis of Zeolites. *Chem. Rev.* **114**, 1521–1543 (2014).
21. Cao, G., Afeworki, M., Kennedy, G. J., Strohmaier, K. G. & Dorset, D. L. Structure of an aluminophosphate EMM-8: a multi-technique approach. *Acta Cryst.* **B63**, 56-62 (2007).
22. Ristic, A., Logar, N. Z., Henninger, S. K. & Kaucic, V. The performance of small-pore microporous aluminophosphates in low-temperature solar energy storage: the structure–property relationship. *Adv. Funct. Mater.* **22**, 1952–1957 (2012).
23. Afeworki, M., Cao, G., Dorset, D. L., Strohmaier, K. G. & Kennedy, G. J. Multinuclear and multidimensional solid-state NMR characterization of EMM-8. *Microporous Mesoporous Mater.* **103**, 216-224 (2007).
24. Canivet, J., Fateeva, A., Guo, Y., Coasne, B. & Farrusseng, D. Water adsorption in MOFs: fundamentals and applications. *Chem. Soc. Rev.* **43**, 5594-5617 (2014).
25. Demontis, P., Gulin-Gonzalez, J., Masia, M. & Suffritti, G. B. The behaviour of water confined in zeolites: molecular dynamics simulations versus experiment. *J. Phys.: Condens. Matter* **22**, 284106 (2010).
26. Bauer, J., Selvam, T., Ofili, J., Che, E., Herrmann, R. & Schwieger, W. Stability of AIPO and SAPO molecular sieves during adsorption-desorption cycles of water vapor investigated by in-situ XRD measurements. *Stud Surf Sci Catal.* **107**, 837–844 (2007).
27. Fischer, M. Interaction of water with (silico)aluminophosphate zeotypes: a comparative investigation using dispersion-corrected DFT. *Phys. Chem. Chem. Phys.* **18**, 15738–15750 (2016).
28. Ristić, A., Krajnc, A. & Logar, N. Z. New Water Adsorbent for Adsorption Driven Chillers. In *Proceedings of the ISES Eurosun 2018 conference-12th International Conference on Solar Energy for Buildings and Industry*, 704-709 (Intal Solar. Energy. Soc. Press, 2018).

REVIEWERS' COMMENTS

Reviewer #1 (Remarks to the Author):

The authors have devoted much time and effort to judiciously address the earlier concerns and queries of this reviewer. The manuscript is now ready for acceptance for potential publication.

Reviewer #2 (Remarks to the Author):

The responses provided by the authors addresses all my comments. However, not all the information was added to the text. I do not see a value of the discussion in the response letter, if it does not propagate to the manuscript. Please extend the changes you made with regard to comment 1 in the manuscript itself.

The resolution of the figures looks still low. However, it could be due to file conversion, and I leave it to the technical editors of the journal.

Reviewer #3 (Remarks to the Author):

The authors' response to the reviewers' comments and suggestions is in-depth and extensive. I think that they properly addressed most of the exposed issues, at least the ones from my side. As I already mentioned in the first review, the materials hold a promise for the selected application. The only thing that I am still not absolutely convinced about is the claim of the "low-cost" synthesis, if all parameters (not only cost of raw materials) are considered. But as explained by the authors, it is a quite relative category and the low-cost and green approaches for aluminophosphate synthesis are intensively studied.

Response to the Reviewer' Comments

Reviewer 1: The authors have devoted much time and effort to judiciously address the earlier concerns and queries of this reviewer. The manuscript is now ready for acceptance for potential publication.

Reply: We would like to thank the reviewer for your recommendation.

Reviewer 2: The responses provided by the authors addresses all my comments. However, not all the information was added to the text. I do not see a value of the discussion in the response letter, if it does not propagate to the manuscript. Please extend the changes you made with regard to comment 1 in the manuscript itself.

The resolution of the figures looks still low. However, it could be due to file conversion, and I leave it to the technical editors of the journal.

Reply: The authors would like to thank the reviewer for carefully evaluating the manuscript. We have added the more detailed description in the revised manuscript to address the comment 1. All Figures are revised and provided as individual vector files according to the guidance of editor, which may look clearer.

Changes add to manuscript:

P19: Although the autoclave synthesis in fluoride-containing medium in this work may be a limitation of this material, methods of ionothermal synthesis, reuse of the organic templates can be employed to synthesize this sorbent in more environment-friendly way.

P20: The aluminum precursor, 0.662 g pseudo-boehmite, was slowly added to a mixture solution of 3.6 ml deionized water and 0.682 ml 85% phosphoric acid. The solution was strongly stirred at room temperature for 10 min to form a homogeneous suspension. Then, 0.112 ml of mineralizer 40% HF and 1.234g 4-dimethylaminopyridine were added to the mixture. After another 10min of stirring and 10 min of ultrasonication, the gel was transferred to a 50 ml Teflon-lined stainless-steel autoclave. The reactor was heated and kept at 175°C for 72~84h under static conditions. After cooling down to room temperature, the white powder of EMM-8 was collected by centrifugation, washed with deionized water, and dried in a vacuum oven. The synthesized samples were activated by calcination at 600 °C for 6 h with heating rates of 5 °C•min⁻¹ to remove the SDA. Three samples obtained by repetitive fabrication progresses were used to demonstrate the consistency of the synthesis, as shown in Supplementary Fig. 20. After water adsorption, this AlPOs is regenerated at temperature above 65 °C.

Reviewer 3:

The authors' response to the reviewers' comments and suggestions is in-depth and extensive. I think that they properly addressed most of the exposed issues, at least the ones from my side. As I already mentioned in the first review, the materials hold a promise for the selected application. The only thing that I am still not absolutely convinced about is the claim of the "low-cost"

synthesis, if all parameters (not only cost of raw materials) are considered. But as explained by the authors, it is a quite relative category and the low-cost and green approaches for aluminophosphate synthesis are intensively studied.

Reply: The authors would like to thank the reviewer for carefully evaluating the manuscript and for giving positive comments on our work. According to the reviewer's suggestion, we remove some claims of low-cost synthesis in the revised manuscript. Only statement about low cost of raw materials retains based on the results in Supplementary Table 12. And the potential of the low-cost and green synthesis of aluminophosphate is discussed.

Changes add to manuscript:

Abstract: Here, we propose a zeolite-like aluminophosphate with SFO topology (EMM-8) for water-sorption-driven refrigeration.

P19: Although the autoclave synthesis in fluoride-containing medium in this work may be a limitation of this material, methods of ionothermal synthesis, reuse of the organic templates can be employed to synthesize this sorbent in more cost-effective and environment-friendly way.

P20: The developed EMM-8 is expected to be a promising candidate for adsorption-based refrigeration, featuring excellent working performance and desirable stability, combined with unique advantages of scalable synthesis.